

# Multi-proxy assessment of brachiopod shell calcite as a potential archive of seawater temperature and oxygen isotope composition

Thomas Letulle[1], Danièle Gaspard[2], Mathieu Daëron[3], Florent Arnaud-Godet[1], Arnauld Vinçon-Laugier[1], Guillaume Suan[1], Christophe Lécuyer[1].

[1] Univ Lyon, UCBL, ENSL, UJM, CNRS, LGL-TPE, F-69622, Villeurbanne, France

[2] UMR 7207, Centre de Recherche en Paléontologie, Paris (CR2P), CNRS, MNHN, Sorbonne- Université. Muséum national d'Histoire naturelle, 8 Rue Buffon, CP 38, 75005 Paris, France.

[3] Laboratoire des Sciences du Climat et de l'Environnement, LSCE/IPSL, CEA-CNRS-UVSQ, Université Paris-Saclay, Orme des Merisiers, F-91191 Gif-sur-Yvette Cedex, France.

*Correspondence to*: Thomas Letulle (thomas.letulle@univ-lyon1.fr)

**Abstract**

Most of our knowledge of past ocean temperature history is based on the $\delta^{18}O$ measurements of calcium carbonate fossil shells. However, the determination of past temperature using this proxy requires the knowledge of past ocean

$\delta^{18}O$, which is generally poorly constrained. Other carbonate-based paleothermometers, such as Mg/Ca ratios, and clumped isotopes ($\Delta_{47}$), have been developed to estimate independently paleotemperatures, and allow past ocean $\delta^{18}O$ to be calculated using various groups of calcifying organisms. Articulated brachiopods are some of the most commonly used in studies of past oceanic geochemistry and temperature. They are abundant in the fossil record since the Cambrian, and their low Mg-calcite mineralogy has been considered relatively resistant to diagenetic

alteration for decades. Here, we investigate the potential of brachiopod shells as recorders of growing temperature and seawater $\delta^{18}O$ using new brachiopod shell geochemical data, by testing multiple established or supposed carbonate-based paleothermometers.

Modern articulated brachiopod shells covering a wide range of temperatures (-1.9 to 25.5°C), depths (5 to 3431 m) and salinities (33.4 to 37.0 PSU), were analysed for their stable isotope compositions ($\delta^{13}C$, $\delta^{18}O$ and $\Delta_{47}$), and

elemental ratios (Mg/Ca, Sr/Ca, Na/Ca and Li/Ca). Our data allowed us to propose a revised oxygen isotope fractionation equation between modern brachiopod shell calcite and seawater:

$$T = -5.2\ (\pm 0.3)\ (\delta^{18}O_c - \delta^{18}O_{sw}) + 19.9\ (\pm 0.8) \qquad R^2 = 0.95\ (n=53)$$

Where $\delta^{18}O_c$ is in ‰VPDB, $\delta^{18}O_{sw}$ is in ‰VSMOW and T is in °C. The measured $\Delta_{47}$ values show a strong correlation with growing temperatures, but is significantly offset relative to the canonical relationship established

for other biogenic and abiogenic calcium carbonate minerals. Our results strongly support the use of clumped isotopes as an alternative temperature proxy and indicate that brachiopod $\Delta_{47}$ values can be used with $\delta^{18}O$ to estimate past $\delta^{18}O_{sw}$ with a precision of about ±1‰ VSMOW. The obtained Mg/Ca ratios show no relationship with temperatures, indicating that this ratio is a poor recorder of past changes in temperatures, an observation at variance with several previous studies. Brachiopod shell Sr/Ca, Na/Ca and Li/Ca display relatively good and

significant correlations with brachiopod living temperature, but data indicate the influence of environmental and biological factors unrelated to temperature. Our proposed revision of marine temperature and water $\delta^{18}O$ proxies based on brachiopod shell geochemistry is promising to refine the record of these oceanic parameters in the Phanerozoic.



## 1. Introduction

Oxygen isotope ratios ($\delta^{18}$O) of brachiopod shell calcite constitute the most extensive record of marine temperatures over the Phanerozoic eon (Prokoph et al., 2008; Veizer and Prokoph, 2015). Nevertheless, our limited knowledge of $\delta^{18}$O of past seas and oceans prevent confident interpretation of this record, as carbonate $\delta^{18}$O ($\delta^{18}O_c$) is dependent on both growth temperature and growing water $\delta^{18}$O ($\delta^{18}O_{sw}$) (Epstein et al., 1953; Kim and O'Neil, 1997; Kim et al., 2007; Brand et al., 2013, 2019). Other paleotemperature proxies, such as trace element ratios and

clumped isotopes ($\Delta_{47}$), may be the key to build a more confident record of Phanerozoic ocean temperatures. Owing to the determination over the last decades of reliable oxygen isotope fractionation equations between calcite and seawater, such independent temperature estimates can be combined with $\delta^{18}O_c$ measurements to reconstruct $\delta^{18}O_{sw}$ values and hence characterise ocean or local hydrography.

Alternative carbonate thermometers have been developed for different calcifying organisms. Mg/Ca thermometry

of foraminiferal calcite (Nürnberg, 1995; Anand et al., 2003) is a widely used proxy that is the basis for reconstruction of changes in $\delta^{18}O_{sw}$ and ice volumes over the Cenozoic era (Lear, 2000; Billups and Schrag, 2003; Miller et al., 2020). Building on the pioneering work of Lowenstam (1961), Brand et al., (2013, 2019) documented a positive correlation between seawater temperature and the $MgCO_3$ content of brachiopod shells sampled at a worldwide scale, and, following conclusions of Jiménez-López et al. (2004), developed an isotopic fractionation

equation considering the amount of $Mg^{2+}$ in substitution to $Ca^{2+}$ in the crystal lattice of the low-Mg calcitic shell (Brand et al., 2013). This $MgCO_3$ corrected isotopic fractionation equation was abandoned in their more recent works (Brand et al., 2019). $MgCO_3$-temperature relationship is described by a curve named the "global brachiopod Mg line" (GBMgL), yet Brand et al., (2013, 2019) acknowledge species-specific deviations from this concept falling above the "line". Brachiopod shell Mg content also shows a strong taxonomic control with shells of

thecideid and craniid being made of high-Mg calcite, in contrast with the low-Mg calcite of rhynchonellide and terebratulide brachiopod shells (Brand et al., 2003; Ullmann et al., 2017).

Sr/Ca ratios of brachiopod calcite were also suggested as marine paleothermometers owing to their correlations with oxygen isotope ratios and seawater temperature (Lowenstam, 1961; Mii and Grossman, 1994; Pérez-Huerta et al., 2008; Butler et al., 2015; Ullmann et al., 2017), This elemental ratio, initially used to assess the shell

preservation of fossil specimens (Brand and Veizer, 1980), also constitutes a marine paleothermometer in other groups such as corals (McCulloch et al., 1994; Shen et al., 1996; Marshall and McCulloch, 2002; Swart et al., 2002; Ayling et al., 2006; DeLong et al., 2011).

Brachiopod Li/Ca ratio was also suggested as a potential temperature proxy in the seminal study of Delaney et al. (1989). Subsequent studies have also confirmed that Li/Ca may constitute a potentially accurate thermometer in

brachiopods (Dellinger et al., 2018; Rollion-Bard et al., 2019; Washington et al., 2020) and in corals (Montagna et al., 2014; Marchitto et al., 2018). For this latter group, however, Li/Mg ratios have been considered as a much more precise thermometer than Li/Ca (Montagna et al., 2014; Marchitto et al., 2018), despite some identified caveats such as the scrambling role of organic matter (Cuny-Guirriec et al., 2019).

Clumped isotope ($\Delta_{47}$) thermometry has been developed on different carbonate materials (Ghosh et al., 2006;

Zaarur et al., 2013). This parameter measures the anomaly of $^{13}$C-$^{18}$O bonds within the carbonate lattice relative to their stochastic abundance. $\Delta_{47}$ values are strongly correlated to crystallisation temperature and independent



from $\delta^{18}O_{sw}$. This proxy has been applied to different marine calcifying organisms to estimate $\delta^{18}O_{sw}$ in deep past (Petersen et al., 2016; Bergmann et al., 2018; Henkes et al., 2018; Wierzbowski et al., 2018; Price et al., 2020; Vickers et al., 2020, 2021; de Winter et al., 2021; Meckler et al., 2022). Although promising, brachiopod shell $\Delta_{47}$

composition was only used in a couple of studies to estimate past temperatures (Henkes et al., 2013; Came et al., 2014). Indeed, the few existing modern brachiopod $\Delta_{47}$−temperature data are in good agreement within the canonical $\Delta_{47}$−temperature relationship established for other biogenic and abiogenic calcium carbonate minerals (Henkes et al., 2013; Came et al., 2014). Recent compilation of $\Delta_{47}$−T calibration data from different laboratories using the new carbonate-based standardisation (Bernasconi et al., 2018, 2021) concluded to a unique $\Delta_{47}$-T

relationship for synthetic and natural carbonates (Anderson et al., 2021). The establishment of such unique calibration equation strengthens the good agreement observed between previous natural and synthetic $\Delta_{47}$−T relationships and would suggests that this calibration is valid when applied to brachiopod shells. However, $\Delta_{47}$ measurements from brachiopod shells are absent from this compilation, and other studies suggest that they may be subjected to significant deviations from $\Delta_{47}$−T relationships (Bajnai et al., 2018, 2020).

Numerous studies highlighted that brachiopod shell calcite does not precipitate *sensu stricto* in isotopic equilibrium with water but as best mimic thermodynamic equilibrium (Carpenter and Lohmann, 1995; Auclair et al., 2003; Parkinson et al., 2005; Yamamoto et al., 2010a, b; Cusack et al., 2012; Takayanagi et al., 2012, 2013, 2015; Bajnai et al., 2018, 2020; Romanin et al., 2018; Rollion-Bard et al., 2019). Based on the conclusions of Bajnai et al., (2018), Brand et al., (2019) characterise their oxygen isotope fractionation database as representing a "brachiopod

based-equilibrium". This term is opposed to the thermodynamic equilibrium, which is best described by very slow growing calcites (Coplen, 2007; Daëron et al., 2019), thus ensuring isotopic equilibration of the DIC species with water (Watkins et al., 2013, 2014). Trace element incorporation into calcite is also affected by kinetic effects. In synthetic carbonates, the relative abundance of elements such as Sr, Na and Li rise with higher growth rates (Lorens, 1981; Busenberg and Plummer, 1985; Tesoriero and Pankow, 1996; Gabitov et al., 2011, 2014). Such

kinetic trends were evidenced at an intra-individual level among brachiopods as documented by the negative correlation of element/Ca ratios with stable isotope values ($\delta^{13}C$, $\delta^{18}O$) (Ullmann et al., 2017; Rollion-Bard et al., 2019).

In this study, we assess the potential of various geochemical proxies ($\delta^{18}O$; $\Delta_{47}$; Sr/Ca; Mg/Ca, Li/Ca, Na/Ca) as recorders of seawater temperature by analysing a new set of modern articulated brachiopod shells collected from

various depths and latitudes during institutional oceanographic cruises, and covering a broad range of water temperatures comprised between -1.9°C and 25.5°C. We discuss the validity and robustness of our oxygen isotope fractionation equation established with measured $\delta^{18}O_c$ of calcite, and seawater temperature and $\delta^{18}O_w$ estimated from oceanographic data. The dependence of elemental ratios to seawater temperature is discussed at the light of this new available dataset of modern brachiopods. Finally, we highlight kinetic effects registered in the dataset of

brachiopod shell calcite and we provide suggestions on how they may be considered for proxy data interpretation.

## 2. Material and Methods

### 2.1 Sample collection and environmental parameters



The studied material consists of 37 articulated brachiopod shells that were collected *in situ* during institutional
oceanographic cruises or other scientific missions (see supplementary data for detail). Sampling location was
documented with depth and geographic coordinates for most samples. Temperature and salinity, which were not
measured *in situ* for most samples, were estimated independently using the NOAA World Ocean Atlas 2018
(Locarnini et al., 2018; Zweng et al., 2018) or, whenever available, using local records in the literature (Table 1).
For both temperature and salinity, mean annual values (MAT), as well as higher and lower monthly averages, were
considered. Similarly, $\delta^{18}O_{sw}$ values were usually not measured *in situ*, but calculated from the salinity data with
the appropriate regional $\delta^{18}O_{sw}$-S relationship from LeGrande and Schmidt (2006). The oceanographic parameters
used in this study are listed in Table 1.

### 2.2 Sample preparation

Encrusting organisms covering the shells were first mechanically removed using stainless steel dental tools and
each shell was then placed in diluted bleach (NaClO 5 %), for 5 to 10 min in an ultrasonic bath to remove organic
matter and other contaminants. As preliminary tests have revealed that the more delicate shells (ex: *Macandrevia
africana* Cooper, 1975) broke apart during the ultrasonic bath, immersion in diluted bleach for few hours without
ultrasonic bath was subsequently preferred for few specimens. In all cases, the specimens were rinsed with
deionized water and oven-dried at 50°C for a few hours.

The sampling of brachiopod shell calcite was performed following the recommendations of Romanin et al (2018).
The umbo, edges and the muscle scar area were avoided because they record major kinetic or metabolic effects
(Carpenter and Lohmann, 1995; Auclair et al., 2003; Parkinson et al., 2005; Yamamoto et al., 2010b; Ullmann et
al., 2017; Romanin et al., 2018), and sampling was focused on the middle part of the shell. Considering the complex
structure of brachiopod shells (Simonet Roda et al., 2022), the unambiguous distinctions between the different
shell layers could not be performed during sampling. In this study, we differentiate the outer layers, i.e. a mix of
the primary layer and some amount of the outer secondary fibrous layer which record significant kinetic effects
(Carpenter and Lohmann, 1995; Auclair et al., 2003; Parkinson et al., 2005; Bajnai et al., 2018; Romanin et al.,
2018; Rollion-Bard et al., 2019), from the inner layers, i.e. the inner secondary fibrous layer and/or the tertiary
prismatic layer when present. Outer shell layers were removed manually with dental tools and resulting powders
were kept to compare their elemental ratios, $\delta^{18}O$ and $\delta^{13}C$ with the inner shell layers (Carpenter and Lohmann,
1995; Auclair et al., 2003; Pérez-Huerta et al., 2008). After removal of the outer layers, the inner layers were
sampled using a graving bit fitted to Dremel Micro™ drill adjusted to the lowest possible speed. For each sample,
the powder was used for all selected geochemical analyses which correspond to $\delta^{18}O$, $\delta^{13}C$, $\Delta_{47}$, and element
concentration. As clumped isotope analyses require large amounts of material (>10 mg are necessary to operate
multiple replicate analyses), a large area (a few cm²) was sampled in the middle part of the shell, allowing several
mg of calcitic powder to be collected. It is worthy to note that this large sampling area may correspond to several
months or years of shell growth depending on the species.

This sampling protocol was applied to most of our samples, except to the smallest and most fragile shells that
quickly broke apart during sampling. Consequently, in the case of the small shells of *Terebratulina latifrons*,
*Frenulina sanguinolenta* and *Fallax neocaledonensis*, the outer layers were kept and only the area covering the
umbo up to the muscle scar was avoided for sampling. The weakness and thinness of *M. africana* precluded a total





**Table 1:** Brachiopod samples with taxonomic identification, sampling location and environmental parameters. Most environmental parameters are derived from the World Ocean Atlas 2018 (Locarnini et al., 2018; Zweng et al., 2018). For a few samples, environmental parameters are derived from local literature as listed below. Almost all $\delta^{18}O_{sw}$ values are calculated using regional $\delta^{18}O_{sw}$–Salinity relationship published by LeGrande and Schmidt (2006), unless direct measurements covering seasonal variations was available in the literature.

1. Goodwin and Cornelisen, (2012); 2. Woods et al. (2014); 3. Meredith et al. (2013); 4. Jacobson, (1983)

| Specimen | Taxa | Living location | Depth (m) | MAT (°C) | Seasonal T variation (°C) | Salinity | Seasonal salinity | $\delta^{18}O_{sw}$ (‰ VSMOW) |
|---|---|---|---|---|---|---|---|---|
| | | New Caledonia | | | | | | |
| FNEO-N4 | *Fallax neocaledonensis* | 22°54'S - 167°13'E | 403-429 | 13.4 | ±1.2 | 35.11 | ±0.05 | 0.60 |
| FNEO-M2 | *Fallax neocaledonensis* | 18°46'S - 163°16'E | 600 | 7.7 | ±0.6 | 34.55 | ±0.03 | 0.45 |
| FSAN-3 | *Frenulina sanguinolenta* | 20°52'S - 167°08'E | 05-20 | 25.4 | ±1.8 | 35.18 | ±0.10 | 0.62 |
| SCRO-1 | *Stenosarina crosnieri* | 22°59'S - 167°19'E | 525 | 10.0 | ±0.9 | 34.76 | ±0.01 | 0.51 |
| SCRO-3 | *Stenosarina crosnieri* | 22°59'S - 167°19'E | 525 | 10.0 | ±0.9 | 34.76 | ±0.01 | 0.51 |
| SGLO-S1 | *Stenosarina globosa* | 19°06'S - 163°30'E | 215-225 | 19.7 | ±0.7 | 35.61 | ±0.01 | 0.73 |
| SGLO-M1 | *Stenosarina globosa* | 18°59'S - 163°24'E | 320 | 16.4 | ±0.7 | 35.40 | ±0.02 | 0.68 |
| SGLO-M2 | *Stenosarina globosa* | 18°59'S - 163°24'E | 320 | 16.4 | ±0.7 | 35.40 | ±0.02 | 0.68 |
| | | Guadeloupe | | | | | | |
| TGAL-3 | *Tichosina cubensis* | 15°53'N - 61°25'W | 262-266 | 16.6 | ±1.0 | 36.22 | ±0.04 | 0.82 |
| TGAL-4 | *Tichosina cubensis* | 15°53'N - 61°25'W | 262-266 | 16.6 | ±1.0 | 36.22 | ±0.04 | 0.82 |
| TCUB-2 | *Tichosina cubensis* | 16°20'N - 60°57'W | 250 | 17.3 | ±1.2 | 36.38 | ±0.03 | 0.85 |
| TCUB-3 | *Tichosina cubensis* | 16°20'N - 60°57'W | 250 | 17.3 | ±1.2 | 36.38 | ±0.03 | 0.85 |
| TPLI-2 | *Tichosina cf. plicata* | 16°21'N - 60°54'W | 111-162 | 23.6±2 | ±1.0 | 36.8±0.1 | ±0.15 | 0.92 |
| TPLI-5 | *Tichosina cf. plicata* | 16°21'N - 60°54'W | 111-162 | 23.6±2 | ±1.0 | 36.8±0.1 | ±0.15 | 0.92 |
| TDES-G1 | *Tichosina sp.* | 16°21'N - 60°54'W | 111-162 | 23.6±2 | ±1.0 | 36.8±0.1 | ±0.15 | 0.92 |
| TDES-G3 | *Tichosina sp.* | 16°21'N - 60°54'W | 111-162 | 23.6±2 | ±1.0 | 36.8±0.1 | ±0.15 | 0.92 |
| TDES-B2 | *Tichosina cf. cubensis* | 16°21'N - 60°54'W | 111-162 | 23.6±2 | ±1.0 | 36.8±0.1 | ±0.15 | 0.92 |
| TDES-B4 | *Tichosina cf. cubensis* | 16°21'N - 60°54'W | 111-162 | 23.6±2 | ±1.0 | 36.8±0.1 | ±0.15 | 0.92 |
| TLAT-5 | *Terebratulina latifrons*[1] | 16°21'N - 60°54'W | 111-162 | 23.6±2 | ±1.0 | 36.8±0.1 | ±0.15 | 0.92 |





| Specimen | Taxa | Living location | Depth (m) | MAT (°C) | Seasonal T variation (°C) | Salinity | Seasonal salinity | $\delta^{18}O_{sw}$ (‰ VSMOW) |
|---|---|---|---|---|---|---|---|---|
| | | **New Zealand** | | | | | | |
| WB5[1] | *Terbratella sanguinea* | 45°20.86'S - 167°02.86'E | 12-20 | 13.6 | +3.3; -1.7 | 34.70 | | 0.33 |
| WB6[1] | *Liothyrella neozelanica* | 45°19'30''S - 166°59'24''E | 20-30 | 13.2 | ±2.5 | 34.70 | | 0.33 |
| WB8 | *Notosaria nigricans* | 45°21'36''S - 170°50'24''E | 20 | 10.4 | ±2.0 | 34.34 | ±0.05 | 0.16 |
| WB9A[2] | *Calloria incospicua* | 43°34'27''S - 172°40'07''E | 20 | 14.0 | ±4 | 33.40 | ±0.90 | -0.26 |
| | | **Crozet Islands** | | | | | | |
| WB4A | *Aerothyris kerguelenensis* | 45°57'36''S - 50°03'24''E | 200 | 3.9 | ±0.6 | 33.97 | ±0.02 | -0.30 |
| WB4B | *Aerothyris kerguelenensis* | 46°06'0''S - 50°38'18''E | 212-230 | 3.7 | ±0.6 | 34.01 | ±0.03 | -0.29 |
| AKER-52 | *Aerothyris kerguelenensis* | 45°48'06''S - 49°45'45''E | 355 | 3.1 | ±0.3 | 34.16 | ±0.01 | -0.25 |
| AKER-66 | *Aerothyris kerguelenensis* | 46°40'00''S - 51°40'30''E | 325 | 2.8 | ±0.6 | 34.16 | ±0.01 | -0.25 |
| AKER-12 | *Aerothyris kerguelenensis* | 46°07'24''S - 50°46'18''E | 290-305 | 3.2 | ±0.2 | 34.11 | ±0.01 | -0.26 |
| AKER-73 | *Aerothyris kerguelenensis* | 46°28'30''S - 51°35'00''E | 207-215 | 3.3 | ±0.8 | 34.00 | ±0.02 | -0.29 |
| AKER-68 | *Aerothyris kerguelenensis* | 46°32'54''S - 51°47'00''E | 200 | 3.4 | ±0.8 | 33.99 | ±0.03 | -0.29 |
| AKER-79 | *Aerothyris kerguelenensis* | 45°51'24''S - 50°44'00''E | 140 | 4.2 | ±0.6 | 33.90 | ±0.01 | -0.31 |
| AKER-61 | *Aerothyris kerguelenensis* | 46°28'00''S - 51°53'12'' | 105 | 4.1 | ±1.2 | 33.85 | ±0.01 | -0.33 |
| | | **Antarctica** | | | | | | |
| WB1A | *Magellania fragilis* | 66°38'S - 143°05'E | 862-875 | -1.9 | +0.3 | 34.70 | - | -0.13 |
| MFRA-CEA | *Magellania fragilis* | 66°38'S - 143°05'E | 862-875 | -1.9 | +0.3 | 34.70 | - | -0.13 |
| LUVA-PAL[3] | *Liothyrella uva* | 67°34'S - 68°08'W | 10-30 | -1.1 | +0.4 | 33.30 | ±0.20 | -0.65 |
| | | **Norway** | | | | | | |
| MCRA-SKA[4] | *Macandrevia cranium* | 63°52'N - 11°04'E | 40-100 | 7 | ±1 | 33.5 | ±0.5 | -0.56 |
| | | **Offshore Angola** | | | | | | |
| MAF-5 | *Macandrevia africana* | 12°21.4'S - 11°02.7'E | 3431 | 2.5 | - | 34.90 | - | 0.05 |



removal of the outer layers, however, the umbo, muscle scars and edges were removed from the bulk shell.
Fragments of these more fragile shells were ground to a fine powder in an agate mortar.

### 2.3 $\delta^{13}C$ and $\delta^{18}O$ of modern brachiopod calcite

Stable isotope compositions of the sampled powders were determined using an auto sampler Multiprep[TM] coupled to a dual-inlet and a GV Isoprime® mass spectrometer. For each sample, an aliquot of about 400 µg of calcium carbonate was reacted with anhydrous oversaturated phosphoric acid at 90 °C during 20 min. Oxygen isotope
ratios of calcium carbonate were computed assuming an acid fractionation factor $1000\ln\alpha(CO_2–CaCO_3)$ of 8.1 between carbon dioxide and calcite (Swart et al., 1991). All sample measurements were duplicated and adjusted to the international references NIST NBS18 ($\delta^{18}O_{VPDB}$ = -23.2‰; $\delta^{13}C_{VPDB}$ = -5.01‰) and NBS19 ($\delta^{18}O_{VPDB}$ = -2.20‰; $\delta^{13}C_{VPDB}$ = +1.95‰), and an internal standard of Carrara marble ($\delta^{18}O_{VPDB}$ = -1.84‰; $\delta^{13}C_{VPDB}$ = +2.03‰). Since 2019 reproducibility of the Carrara Marble in-house standard has been ±0.089‰ for $\delta^{18}O$ ($2\sigma$,
n=668) and ±0.064‰ for $\delta^{13}C$ ($2\sigma$, n=668). These stable isotope data are completed with $\delta^{18}O$ and $\delta^{13}C$ data obtained from clumped isotope measurements for which the analytical procedures are reported in detail in section 2.5.

### 2.4 Elemental ratios

Elemental concentrations were obtained by dissolving 2 to 20 mg of carbonate powder in 10 ml $HNO_3$ (2%). For all samples, pairs of aliquots were prepared with 10 times and 100 times dilutions depending on the considered trace and major elements along with a fixed amount of Sc and In that were added to correct concentrations from instrument drift. Solutions were analysed using an Inductively Coupled Plasma-Optical Emission Spectrometer (iCAP 7000 ICP-OES) and a quadrupole ICP-Mass Spectrometer (i-CAP-Q ICP-MS), for minor and trace
elements, respectively. The 100x diluted aliquots were used to calculate Ca concentration, while 10x diluted aliquots served for Mg, Sr, Na and Li concentrations. Element/Ca ratios were thus calculated and are reported in mmol/mol for Mg/Ca, Sr/Ca and Na/Ca, and in µmol/mol for Li/Ca. The reproducibility of measurements was assessed through the analysis of the carbonate standard CCH1 (Roelandts and Duchesne, 1988).

### 2.5 Clumped isotopes

Carbonate samples were converted to $CO_2$ by phosphoric acid reaction at 90 °C in a common, stirred acid bath for 15 minutes. Initial phosphoric acid concentration was 103 % (1.91 g/cm³) and each batch of acid was used for 7 days. After cryogenic removal of water, the evolved $CO_2$ was helium-flushed at 25 ml/min through a purification column packed with Porapak Q (50/80 mesh, 1 m length, 2.1 mm ID) and held at −20°C, then quantitatively
recollected by cryogenic trapping and transferred into an Isoprime 100[TM] dual-inlet mass spectrometer equipped with six Faraday collectors (m/z 44–49). Each analysis took about 2.5 hours, during which analyte gas and working reference gas were allowed to flow from matching, 10 ml reservoirs into the source through deactivated fused silica capillaries (65 cm length, 110 µm ID). Every 20 minutes, gas pressures were adjusted to achieve m/z = 44 current of 80 nA, with differences between analyte gas and working gas generally below 0.1 nA. Pressure-



dependent background current corrections were measured 12 times for each analysis. All background measurements from a given session were then used to determine a mass-specific relationship linking background intensity ($Zm$), total m/z = 44 intensity ($I44$), and time ($t$):

$Zm = aI_{44} + P(t)$, with $P$ being a polynomial of degree 2 to 4.

Background-corrected ion current ratios (δ45 to δ49) were converted to $\delta^{13}C$, $\delta^{18}O$, and "raw" $\Delta_{47}$ values as
described by Daëron et al., (2016), using the IUPAC oxygen-17 correction parameters. The isotopic composition ($\delta^{13}C$, $\delta^{18}O$) of our working reference gas was computed based on the nominal isotopic composition of carbonate standard ETH-3 (Bernasconi et al., 2018) and an oxygen-18 acid fractionation factor of 1.00813 (Kim et al., 2007). Raw $\Delta_{47}$ values were then converted to the I-CDES $\Delta_{47}$ reference frame by comparison with four "ETH" carbonate standards (ETH 1-4, Bernasconi et al., 2021) using a pooled regression approach (Daëron, 2021). Full analytical
errors are derived from the external reproducibility of unknowns and standards (Nf = 89) and conservatively account for the uncertainties in raw $\Delta_{47}$ measurements as well as those associated with the conversion to the "absolute" $\Delta_{47}$ reference frame.

### 3. Results

All geochemical results obtained from modern brachiopods ($\delta^{13}C$, $\delta^{18}O$, $\Delta_{47}$, Mg/Ca, Sr/Ca, Na/Ca and Li/Ca) are reported in Table 2. Linear regression models of $\Delta^{18}O_{c-w}$, $\Delta_{47}$, Mg/Ca, Sr/Ca, Na/Ca and Li/Ca with Mean Annual Temperature (MAT) are reported in Figure 1 and Table 3.

#### 3.1 Stable isotopes

The whole dataset of $\delta^{13}C$ and $\delta^{18}O$ values from modern brachiopod shells ranges from −2.24‰ to 3.17‰ and from −1.22‰ to 3.98‰, n=73, respectively. There is a robust negative linear correlation ($R^2 > 0.73$, p-slope < 0.001) between $\Delta^{18}O_{c-w}$ (= $\delta^{18}O_c$ − $\delta^{18}O_w$) and MAT (Figure 1, Table 3). The correlation observed in bulk shell samples is indistinguishable from that in inner layer samples (Figure 1A). On the contrary, the correlation obtained from outer layer samples shows a significant offset with both inner layers and bulk layer samples. This is in line
with the differences between the outer and inner layers observed within the same specimen, which are in most cases above analytical uncertainties and range from −3.72 to −0.19‰ (mean = −1.41±0.40‰; 2σ; n=17) and −0.78 to 0.45‰ (mean = −0.47±0.28‰; 2σ; n=17) for $\delta^{13}C$ and $\delta^{18}O$, respectively.

Our new set of data from inner and bulk layers' samples provides the following oxygen isotope fractionation equation:

T= −5.2(±0.4) $\Delta^{18}O_{c-w}$ + 19.9(±0.8)          $R^2 = 0.95$ (n = 53)

with T the temperature in °C and $\Delta^{18}O_{c-w}$ (= $\delta^{18}O_c$ − $\delta^{18}O_w$) the oxygen isotope fractionation between brachiopod calcite and seawater, excluding data from the outer shell layers.



**Table 2:** Geochemical data of modern brachiopod samples. Stable isotope values and element/Ca ratios.

* Samples with the anterior part of the shell.

| Specimen | Shell layer | δ¹³C (‰ VPDB) | δ¹⁸O_c (‰ VPDB) | Δ₄₇ (‰ I-CDES) | Mg/Ca (mmol/mol) | Sr/Ca (mmol/mol) | Li/Ca (µmol/mol) |
|---|---|---|---|---|---|---|---|
| **New Caledonia** | | | | | | | |
| FNEO-N4 | Bulk | 1.87 | 1.29 | | | | |
| FNEO-N4 | Bulk | 1.93 | 1.32 | | 13.35 | 1.06 | 30.21 |
| FNEO-M2 | Bulk* | 1.93 | 2.06 | | | | |
| FNEO-M2 | Bulk* | 1.97 | 2.05 | | 24.90 | 1.36 | 33.47 |
| FSAN-3 | Bulk* | 1.47 | -1.20 | | | | |
| FSAN-3 | Bulk* | 1.42 | -1.22 | | 14.65 | 1.08 | 22.45 |
| SCRO-1 | Inner | 2.72 | 2.04 | 0.6387 | | | |
| SCRO-3 | Outer | 2.34 | 2.03 | | 13.21 | 1.07 | 28.90 |
| SCRO-3 | Inner | 3.17 | 2.13 | | 3.76 | 0.45 | 4.09 |
| SGLO-S1 | Outer | 2.04 | 0.20 | | 11.15 | 1.02 | 25.53 |
| SGLO-S1 | Inner | 3.06 | 0.44 | | 6.68 | 0.60 | 5.21 |
| SGLO-M1 | Inner | 2.87 | 1.10 | | 6.70 | 0.55 | 6.01 |
| SGLO-M1 | Outer | 1.98 | 0.91 | | 10.72 | 0.94 | 26.91 |
| SGLO-M1 | Inner | 2.99 | 1.12 | | | | |
| SGLO-M2 | Outer | 1.87 | 0.91 | 0.6388 | | | |
| SGLO-M2 | Inner | 2.80 | 0.88 | 0.6210 | | | |
| | | | | | | | |
| **Guadeloupe** | | | | | | | |
| TGAL-3 | Bulk | 1.97 | 1.22 | 0.6312 | | | |
| TGAL-4 | Outer | 1.46 | 1.08 | | 15.56 | 1.12 | 30.04 |
| TGAL-4 | Inner | 2.67 | 1.27 | | 6.87 | 0.53 | 3.89 |
| TGAL-4 | Inner | 2.75 | 1.44 | 0.6113 | | | |
| TCUB-2 | Bulk | 1.75 | 1.08 | 0.6248 | | | |
| TCUB-3 | Outer | 1.45 | 0.98 | | 8.25 | 0.85 | 18.77 |
| TCUB-3 | Inner | 2.31 | 1.24 | | 4.87 | 0.50 | 3.86 |
| TCUB-3 | Inner | 2.25 | 1.25 | | | | |
| TPLI-2 | Inner | 2.75 | 0.14 | 0.5974 | | | |
| TPLI-5 | Inner | 2.96 | 0.45 | | 9.82 | 0.63 | 5.29 |
| TDES-G1 | Inner | 2.68 | 0.18 | 0.6049 | | | |
| TDES-G3 | Inner | 2.76 | 0.31 | | | | |
| TDES-G3 | Inner | 2.61 | 0.23 | | 3.82 | 0.56 | 3.28 |
| TDES-B2 | Inner | 2.58 | 0.26 | 0.6086 | | | |
| TDES-B4 | Inner | 2.56 | 0.36 | | | | |
| TDES-B4 | Inner | 2.71 | 0.47 | | 6.96 | 0.58 | 4.64 |
| TLAT-5 | Bulk* | 1.07 | 0.19 | | | | |
| TLAT-5 | Bulk* | 1.04 | 0.21 | | 25.30 | 1.22 | 28.60 |
| | | | | | | | |
| **New Zealand** | | | | | | | |
| WB5 | Outer | -1.54 | -0.63 | | 6.69 | 1.46 | 40.84 |
| WB5 | Inner | 0.52 | 0.47 | 0.6639 | 4.28 | 1.13 | 38.39 |
| WB6 | Outer | 1.93 | 1.05 | | 12.50 | 1.43 | 35.07 |
| WB6 | Inner | 2.91 | 1.55 | 0.6295 | 11.14 | 0.82 | 15.92 |
| WB8 | Inner | 2.10 | 1.25 | 0.6500 | 11.48 | 1.24 | 46.68 |
| WB9A | Outer | 0.43 | -0.53 | | | | |
| WB9A | Inner | 2.35 | 0.85 | | 5.73 | 1.04 | 34.38 |





| Specimen | Shell layer | $\delta^{13}C$ (‰ VPDB) | $\delta^{18}O_c$ (‰ VPDB) | $\Delta_{47}$ (‰ I-CDES) | Mg/Ca (mmol/mol) | Sr/Ca (mmol/mol) | Li/Ca (µmol/mol) |
|---|---|---|---|---|---|---|---|
| **Crozet Islands** | | | | | | | |
| WB4A | Outer | 1.11 | 2.04 | | 5.59 | 1.18 | 49.33 |
| WB4A | Inner | 2.50 | 2.96 | 0.6664 | 5.13 | 1.00 | 41.10 |
| WB4B | Outer | 1.13 | 2.22 | | 5.57 | 1.20 | 51.41 |
| WB4B | Inner | 1.86 | 2.79 | 0.6752 | 5.08 | 1.01 | 40.80 |
| AKER-52 | Outer | 1.39 | 2.78 | | 8.46 | 1.18 | 48.79 |
| AKER-52 | Inner | 2.03 | 3.37 | 0.6670 | | | |
| AKER-52 | Inner | 2.04 | 3.23 | | 7.47 | 0.94 | 44.70 |
| AKER-66 | Outer | 2.27 | 3.18 | | 9.94 | 1.37 | 48.01 |
| AKER-66 | Inner | 2.46 | 2.73 | | 3.57 | 0.60 | 16.27 |
| AKER-12 | Outer | 0.88 | 2.57 | | 7.46 | 1.25 | 51.65 |
| AKER-12 | Inner | 1.92 | 2.51 | | 3.92 | 0.89 | 38.68 |
| AKER-73 | Outer | 0.58 | 1.96 | | 9.75 | 1.26 | 46.63 |
| AKER-73 | Inner | 1.97 | 2.35 | | 6.80 | 1.08 | 46.41 |
| AKER-68 | Outer | -1.06 | 1.34 | | 5.53 | 1.37 | 51.73 |
| AKER-68 | Inner | 1.57 | 2.57 | 0.6811 | | | |
| AKER-68 | Inner | 1.23 | 2.02 | | 4.92 | 1.06 | 46.71 |
| AKER-79 | Outer | -1.08 | 1.15 | | 6.57 | 1.33 | 49.67 |
| AKER-79 | Inner | 0.54 | 1.33 | | | | |
| AKER-61 | Outer | -2.24 | 0.31 | | 6.09 | 1.55 | 58.47 |
| AKER-61 | Inner | 1.48 | 2.09 | | 4.94 | 1.06 | 48.80 |
| AKER-61 | Inner | 1.36 | 2.26 | 0.6869 | | | |
| | | | | | | | |
| **Antarctica** | | | | | | | |
| MFRA-CEA | Outer | -0.46 | 2.73 | | 5.39 | 1.37 | 56.75 |
| MFRA-CEA | Inner | 1.03 | 3.68 | | 6.84 | 1.13 | 55.35 |
| MFRA-CEA | Inner | 1.01 | 3.98 | 0.6921 | | | |
| WB1A | Bulk | 0.84 | 3.57 | | 5.70 | 1.21 | 36.69 |
| LUVA-PAL | Inner | 0.53 | 2.91 | | 13.47 | 1.56 | 54.82 |
| LUVA-PAL | Inner | 0.63 | 3.33 | 0.6941 | | | |
| LUVA-PAL | Outer | -1.69 | 1.87 | | 20.17 | 2.05 | 56.02 |
| | | | | | | | |
| **Norway** | | | | | | | |
| MCRA-SKA | Outer | -1.32 | 0.54 | | | | |
| MCRA-SKA | Inner | 0.24 | 1.28 | | 7.28 | 1.24 | 53.14 |
| | | | | | | | |
| **Offshore Angola** | | | | | | | |
| MAF-5 | Bulk | 1.67 | 3.56 | | | | |
| MAF-5 | Bulk | 1.57 | 3.48 | | 12.54 | 1.01 | 54.74 |

### 3.2 Trace element/Ca ratios

Mg/Ca, Sr/Ca, Li/Ca and Na/Ca of modern brachiopod shells range from 3.57 to 25.30 mmol/mol, 0.45 to 2.05 mmol/mol, 3.28 to 58.47µmol/mol, and 1.86 to 18.28 mmol/mol, respectively (N=47). Sr/Ca show significant but weak negative correlations ($0.28 < R^2 < 0.44$; p-slope < 0.01) with temperature. Li/Ca show significant and good negative correlations ($0.61 < R^2 < 0.86$; p-slope < 0.001) with temperature, the strongest correlation ($R^2=0.86$) being reached in outer layers samples. Na/Ca show significant negative correlations ($0.38 < R^2 < 0.60$; p-slope < 0.001) with temperature. Concentration of these three elements in brachiopod shells (Sr, Li and Na) are strongly

positively correlated with each other ($0.66 < R^2 < 0.93$ for the whole dataset) especially while considering only





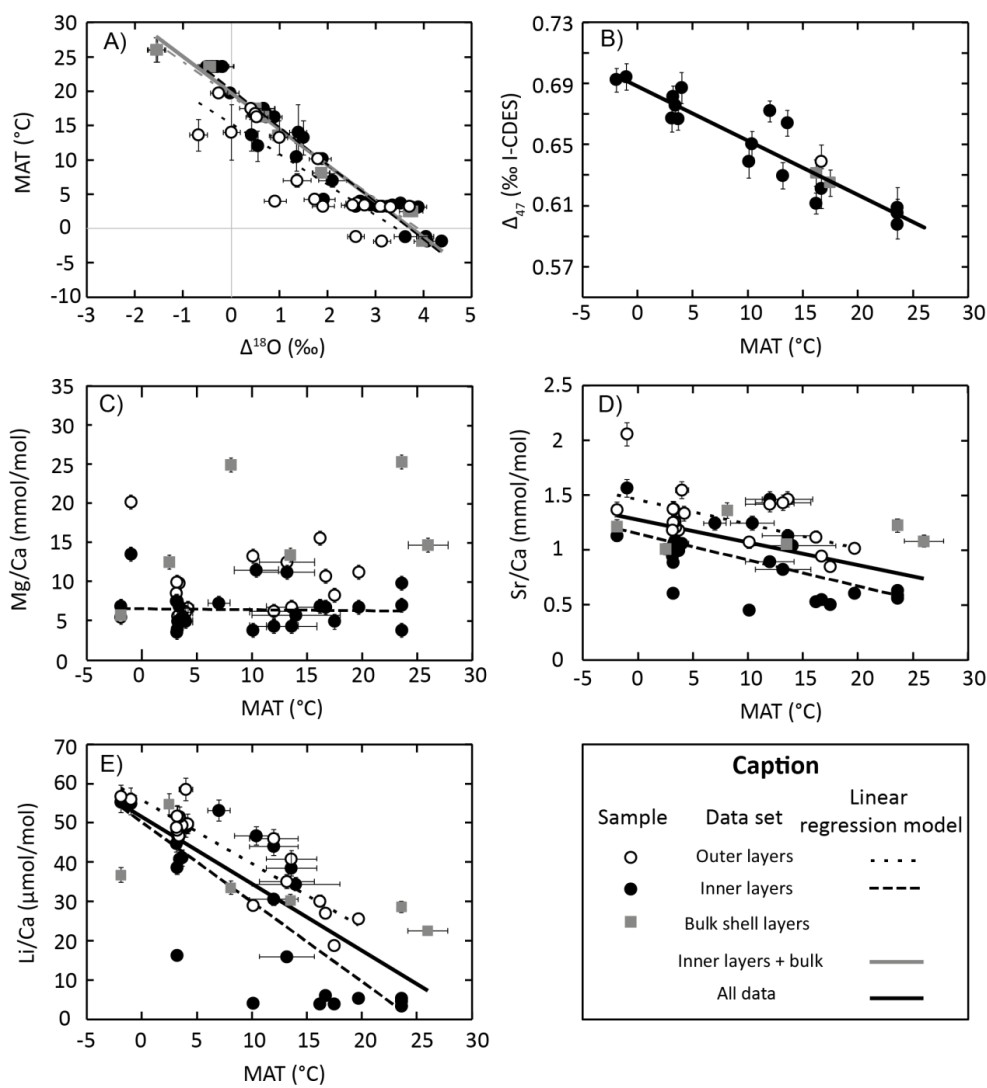

**Figure 1:** Geochemical data and linear regression model for each geochemical parameter tested for its relationship with brachiopod living temperature. Parameters associated with each regression model are listed in table 3.

inner layers samples ($0.84 < R^2 < 0.96$). No significant correlation (p-slope > 0.2) of Mg/Ca with temperature is observed in the dataset (Figure 1; Table 3). Measured element/Ca ratios are systematically higher in the outer layers than in the inner layers of brachiopod shells (Student-t-test: $p < 0.01$; Figure 1C−E).



**Table 3:** Linear regression model parameters for the geochemical parameter-temperature relationship tested and displayed in Figure 1.

| Regression model | Sample/subsample | Nb of points | Slope | Intercept | $\sigma_{slope}$ | $\sigma_{intercept}$ | $\sigma_{residual}$ | P-slope | $R^2$ |
|---|---|---|---|---|---|---|---|---|---|
| MAT (°C) vs $\Delta^{18}O$ (‰) | All data | 74 | -5.1 | 18.7 | 0.2 | 0.5 | 3.0 | <0.001 | 0.87 |
| | Outer layers | 21 | -4.4 | 15.2 | 0.6 | 1.3 | 3.7 | <0.001 | 0.71 |
| | Inner layers | 40 | -5.4 | 20.2 | 0.2 | 0.5 | 2.1 | <0.001 | 0.94 |
| | Bulk sample | 13 | -4.9 | 19.3 | 0.2 | 0.5 | 1.6 | <0.001 | 0.97 |
| | Bulk + inner layers | 53 | -5.2 | 19.9 | 0.2 | 0.4 | 2.0 | <0.001 | 0.95 |
| $\Delta_{47}$ (‰ I-CDES) vs MAT (°C) | All data | 20 | -0.0036 | 0.688 | 0.0003 | 0.005 | 0.011 | <0.001 | 0.86 |
| | Inner layers | 17 | -0.0036 | 0.688 | 0.0004 | 0.005 | 0.013 | <0.001 | 0.86 |
| Mg/Ca (mmol/mol) vs MAT (°C) | All data | 47 | 0.13 | 7.7 | 0.09 | 1.1 | 5.0 | 0.16 | 0.04 |
| | Outer layers | 18 | 0.14 | 8.3 | 0.14 | 1.5 | 4.0 | 0.36 | 0.05 |
| | Inner layers | 23 | -0.0025 | 6.6 | 0.07 | 0.9 | 2.7 | 0.94 | <0.01 |
| Sr/Ca (mmol/mol) vs MAT (°C) | All data | 47 | -0.021 | 1.27 | 0.005 | 0.06 | 0.3 | <0.001 | 0.28 |
| | Outer layers | 18 | -0.022 | 1.45 | 0.007 | 0.08 | 0.2 | 0.008 | 0.36 |
| | Inner layers | 23 | -0.024 | 1.13 | 0.006 | 0.08 | 0.2 | <0.001 | 0.44 |
| Li/Ca (µmol/mol) vs MAT (°C) | All data | 47 | -1.7 | 51.1 | 0.2 | 2.5 | 11 | <0.001 | 0.61 |
| | Outer layers | 18 | -1.7 | 55.6 | 0.2 | 1.7 | 5 | <0.001 | 0.86 |
| | Inner layers | 23 | -2.1 | 49.5 | 0.3 | 4.1 | 12 | <0.001 | 0.67 |
| Na/Ca (mmol/mol) vs MAT (°C) | All data | 47 | -0.32 | 12.7 | 0.06 | 0.7 | 3 | <0.001 | 0.38 |
| | Outer layers | 18 | -0.31 | 14.4 | 0.06 | 0.6 | 2 | <0.001 | 0.60 |
| | Inner layers | 23 | -0.38 | 11.2 | 0.08 | 1.0 | 3 | <0.001 | 0.53 |

### 3.3 Clumped isotopes

$\Delta_{47}$ values obtained from the selected brachiopod shells range from 0.59±0.01 to 0.69±0.01 ‰ I-CDES and are strongly correlated with ambient temperature (Table 3, Figure 1B). Temperatures inferred from clumped isotope data were calculated using the INTERCARB calibration determined by Anderson et al., (2021) who reprocessed the data coming from various laboratories using the same carbonate standards for data correction. Clumped isotope temperatures are significantly lower than estimated MAT with a mean deviation from environmental temperatures
of – 2.4 ± 1.3°C (95% CI; n=19).

### 4. Discussion

#### 4.1 Brachiopod shell paleothermometers.

##### 4.1.1 Validity and robustness of our $^{18}O/^{16}O$-based fractionation equation

The oxygen isotope fractionation equation derived from our data deviates substantially from that also derived from
articulated brachiopod shells determined by Brand et al. (2019) (Figure 2). The two equations follow similar trends and are only offset by ~0.6‰ at tropical temperatures (20-30°C) (Figure 2A) with the equation from Brand et al.





(2019) predicting a lower fractionation factor. At temperate and polar temperatures (20 to 0°C) our equation has a steeper slope than that of Brand et al. (2019). The two equations overlap at around 10°C and deviate substantially when approaching water freezing temperatures (~0°C), with a $\Delta^{18}O_{c-w}$ difference of up to ~1‰ between the

equation of Brand et al, (2019) and our equation (Figure 2 A), a pattern also observed in the geochemical datasets (Figure 2B). Examining the geochemical dataset revealed that the large offset between the two equations at low temperature is partly resulting from the use of a second-order regression model by Brand et al. (2019) as the regression curve overlaps with the highest $\Delta^{18}O_{c-w}$ data points at temperatures of ~1°C. We preferred to use a linear regression more appropriate to fit data in the low temperature (<50°C) linear domain of the hyperbolic curve ($\alpha =$

$f (1/T^2)$) predicted by equilibrium thermodynamics. Still the main discrepancies between the equation proposed herein and that of Brand et al., (2019) is mainly driven by a difference in the geochemical dataset. Those differences may arise from a few reasons, 1) different approaches to determine environmental parameters 2), differences in brachiopod shell preparation and sampling or 3) major differences in the geographical and taxonomical distribution of the brachiopod dataset.


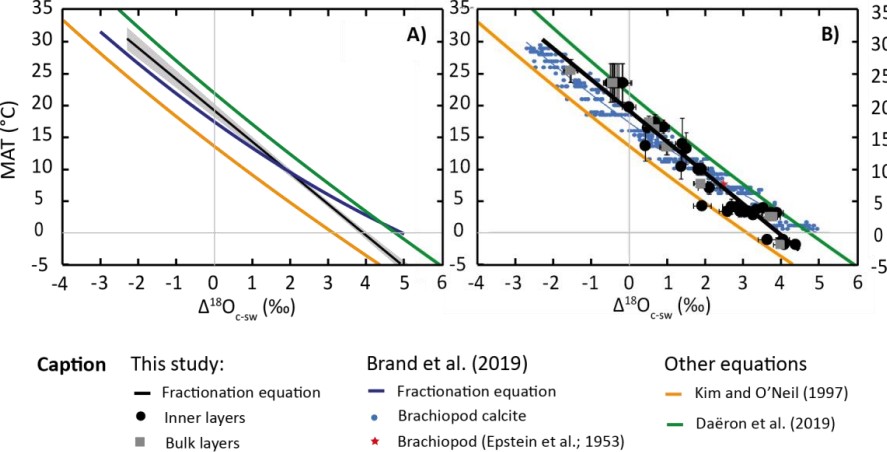

**Figure 2:** Plot comparison of A) regression lines for different temperature-oxygen isotope fractionation for brachiopod calcite (This study, Brand et al. 2019), laboratory precipitated calcite (Kim and O'Neil, 1997) and slow growing cave calcite (Daëron et al., 2019) and B) corresponding datasets for brachiopod calcite (modified after

Brand et al., 2019). Uncertainty envelope 95% CI of the York least square regression (York et al., 2004).

We use a published independent modern brachiopod $\delta^{18}O$ dataset (Bajnai et al., 2018) to test our equation against other previously published equations in their ability to predict environmental temperatures when $\delta^{18}O_{sw}$ is known (Figure 3). This dataset proposes two sets of $\delta^{18}O_{sw}$ for most data. One derived from direct measurements and one

derived from $\delta^{18}O_{sw}$-Salinity relationships (LeGrande and Schmidt, 2006). Plus, Mg/Ca values are available for all samples, allowing us to apply the equation of Brand et al. (2013) to the dataset. Isotopic temperatures were





calculated then normalised as the temperature deviation from environmental temperature (ΔT in °C) with full propagation of uncertainties. Of all the equations tested, the equations derived from measurements of modern brachiopods (Brand et al., 2013, 2019; this study) yield the most accurate temperature predictions. Mean ΔT are

statistically indistinguishable from 0 using salinity-based $\delta^{18}O_{sw}$ estimates result for all three equations.

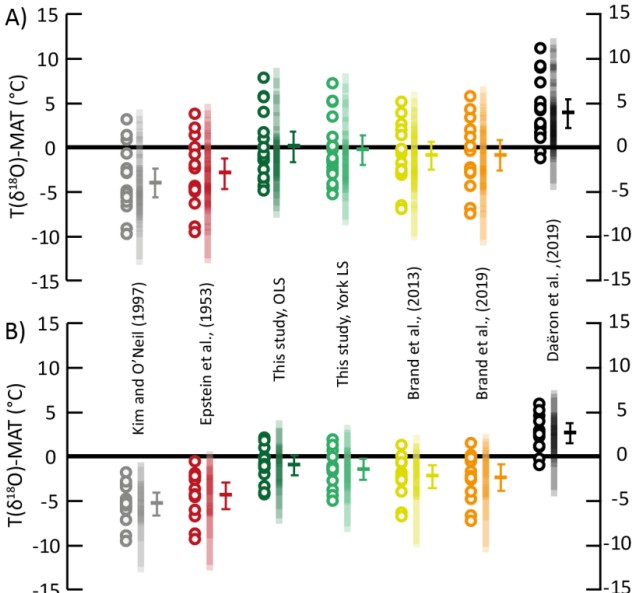

**Figure 3:** Offsets between the δ18O temperatures and MAT from the modern brachiopods dataset of Bajnai et al., (2018) using different fractionation equations. Circles: temperature offset for each samples; vertical transparent

bars: temperature offset 95% confidence interval for each samples; horizontal bar: temperature offset arithmetic mean and associated 95% confidence interval. A) Temperature offset for isotopic temperatures calculated with $\delta^{18}O_{sw}$ values calculated from the LeGrande and Schmidt (2006) database; N=18. B) Temperature offset for isotopic temperatures calculated with locally measured δ18Osw values; N=13. OLS = ordinary least square regression; YLS = York least square regression (York et al., 2004).


Interestingly, while using locally measured $\delta^{18}O_{sw}$ values, the accuracy of isotopic temperature estimates is poorer. The equations of Brand et al. (2013; 2019) now significantly underestimate growing temperatures (Figure 3, $p_{(mean}$ $_{\Delta T = 0)} < 0.01$) while our equation yields mean ΔT that are statistically indistinguishable from 0 ($p_{(mean \Delta T = 0)} = 0.11$). The equation of Kim and O'Neil (1997), derived from synthetic carbonates, and that of Epstein et al. (1953),

derived from molluscs, both underestimate brachiopod shells growing temperature (Figure 3, $p_{(mean \Delta T = 0)} < 0.001$) in both $\delta^{18}O_{sw}$ hypotheses. On the other end of the spectrum, the equation of Daëron et al. (2019), derived from slow growing calcite, overestimates brachiopod shells growing temperatures (Figure 3, $p_{(mean \Delta T = 0)} < 0.001$) in both




$\delta^{18}O_{sw}$ hypotheses. Although it lessens the accuracy of $\delta^{18}O$ temperatures in this dataset, using the locally measured $\delta^{18}O_{sw}$ values over the salinity-based estimates improved their precision ($4.0 < 2\sigma < 5.3$ °C, n=13 rather than 5.4

$< 2\sigma < 6.4$ °C, n=18). Those two $\delta^{18}O_{sw}$ hypotheses regarding $\delta^{18}O_{sw}$ values have their own benefits and drawbacks. $\delta^{18}O_{sw}$ estimated from salinity measurements and the regional $\delta^{18}O_{sw}$-Salinity relationships offers $\delta^{18}O_{sw}$ estimates that can be averaged through time but this approach ignores coastal deviations from the $\delta^{18}O_{sw}$-salinity relationship. Locally measured $\delta^{18}O_{sw}$ correct for those effects, however those measurements are often scarce in time can be can be biased at the seasonal or inter-annual scale for brachiopods living above the base of the thermocline. For

example, Brand et al. (2019) used $\delta^{18}O_{sw}$ values measured in late February (Austral summer) to estimate the isotope fractionation of *Liothyrella uva* from Rothera island, while long term $\delta^{18}O_{sw}$ measurements at this station indicate minimal values during the austral summer (Meredith et al., 2013). Independent from the fractionation equation or the $\delta^{18}O_{sw}$ value used, temperature estimates are associated with large scatter of more than 4°C ($2\sigma$) around mean $\Delta T$. We suggest that this scatter is related primarily to the variability of brachiopod shell $\delta^{18}O$ values observed

within a population (Figure 2, Brand et al., 2019) due to other isotopic fractionation mechanisms than temperature dependent fractionation, namely kinetic effects, metabolic effects and pH effects. Using the $MgCO_3$-corrected equation of Brand et al. (2013) instead of Brand et al. (2019), both derived from very similar datasets, has the only effect of slightly improving the precision of the estimates decreasing the scatter by $\sim 0.6$°C around the mean deviation from environmental temperatures. This effect is about one order of magnitude below the remaining

dispersal of data.

In the following, we assess the impact of trace element incorporation in brachiopod calcite on the fractionation factor between calcite and water using our new data. Brachiopod calcite can be viewed as a three component molar mixture of $CaCO_3$, $MgCO_3$ and $SrCO_3$, and that the fractionation factor of this mixture ($\Delta_{brach.calcite-water}$) of divalent metal carbonates is equivalent to the sum of the mineral-water fractionation factors weighted from the mineral

molar fractions. This bulk isotopic fractionation factor is thus expressed as follows:

$$\Delta_{(brach.calcite-water)} = \sum_{i=1}^{n} X_i . \Delta_{(mineral-water)}$$

With the n (=3) minerals noted i and X the molar fraction of each mineral constituting the brachiopod calcite mixture. As to our knowledge $MgCO_3$-water oxygen isotope fractionation equation has not yet been established from synthetic carbonates, we used the theoretical fractionation factors ($\alpha_{(x-y)}$) of Chacko and Deines (2008) to

compare the mixture and pure calcite fractionation factors. $\Delta_{(brach.calcite-water)}$ calculated for brachiopod inner calcite layers with the same range of $CaCO_3$, $MgCO_3$ and $SrCO_3$ contents similar to that recorded by our elemental data, do not exceed 0.18‰ compared to the calcite end-member, at environmental temperatures (Mean = 0.08‰; N=24). This difference is mainly explained by $Mg^{2+}$ incorporation as 1) $MgCO_3$ is more abundant than $SrCO_3$ in brachiopod calcite and 2) $1000\ln \alpha_{(MgCO3-water)}$ is higher than $1000\ln \alpha_{(CaCO3-water)}$ by about 13‰ in this temperature

range whereas $1000\ln \alpha_{(SrCO3-water)}$ is lower than $1000\ln \alpha_{(CaCO3-water)}$ by only $\sim 1$‰ (Chacko and Deines, 2008). This basic calculation illustrates that $Mg^{2+}$ incorporation into brachiopod low Mg-calcite can affect calcite $\delta^{18}O$ by no more than 0.2‰, which is about twice the range of analytical uncertainties at $2\sigma$. In consequence, the effect of $Mg^{2+}$ has a weak effect on the $\delta^{18}O$ of brachiopod calcite, the $MgCO_3$ correction proposed by Brand et al., (2013) is not really needed for paleoclimatic and paleoceanographic studies, which most likely justifies why it was

later abandoned (Brand et al., 2019).



### 4.1.2 Brachiopod shells clumped-isotopes: an alternative paleothermometer?

Although there is a strong correlation between our new $\Delta_{47}$ values from brachiopod shells and growing temperatures, $\Delta_{47}$ values at a given temperature are generally higher than what is expected from the state-of-the-

art equation of Anderson et al., (2021) (Figure 4). This observation is in line with the results of Bajnai et al., (2018) gathered in the CDES reference frame. To discuss modern brachiopod $\Delta_{47}$ on a larger dataset, the data from Bajnai et al., (2018) were adjusted to the CDES 90 reference frame to better compare them with our data set in the I-CDES reference frame (Bernasconi et al., 2021). We acknowledge that the different standardisation protocols used for the two datasets could hamper the strength of this comparison. However, this comparison is supported by the

good agreement between sample 143 from Bajnai et al., (2018) ($\Delta_{47} = 0.670 \pm 0.008$ ‰ CDES 90) and its replicate sample Mv143b ($\Delta_{47} = 0.664 \pm 0.006$ ‰ I-CDES; Bajnai et al., 2020; Fiebig et al., 2021). Both datasets are first compared to the Anderson et al. (2021) calibration (Figure 4). Many brachiopod shells record higher $\Delta_{47}$ values than what is expected from the calibration of Anderson et al., (2021) (Figure 4A-B), in line with the underestimation $\Delta_{47}$ temperatures relative to MAT when using this relationship (Figure 4C). The temperature- $\Delta_{47}$

relationship derived from the Bajnai et al. (2018) dataset appears significantly different from that of Anderson et al. (2021) equation, the latter being indistinguishable from the relationship derived from our new dataset (Figure 4A). We argue that the difference between the two datasets of modern brachiopod $\Delta_{47}$ may largely results from distinct taxonomic assemblages and/or sampled locations in both datasets. Indeed, our new $\Delta_{47}$ data from New Zealand brachiopods replicate very well with previous data from same brachiopod species and similar locations

(Bajnai et al., 2018) (see supplementary). The $\Delta_{47}$-temperature relationship derived using both brachiopod datasets, is still nearly within error of the equation of Anderson et al., (2021) (Figure 4B). The use of this latter equation to calculate brachiopod growing temperature results in a significant mean offset with environmental temperatures of $\sim -3.9 \pm 1.2°C$ (95% CI, Figure 4C) for the combined brachiopod dataset. In consequence, based on the data available to date, we support the use the following brachiopod equation:

$$\Delta_{47(I-CDES90°C)} = 0.042 \pm 0.002 \times \frac{10^6}{T^2} + 0.128 \pm 0.027$$

with, $\Delta_{47 \, (I\text{-}CDES90°C)}$ in ‰, and T the temperature in K.





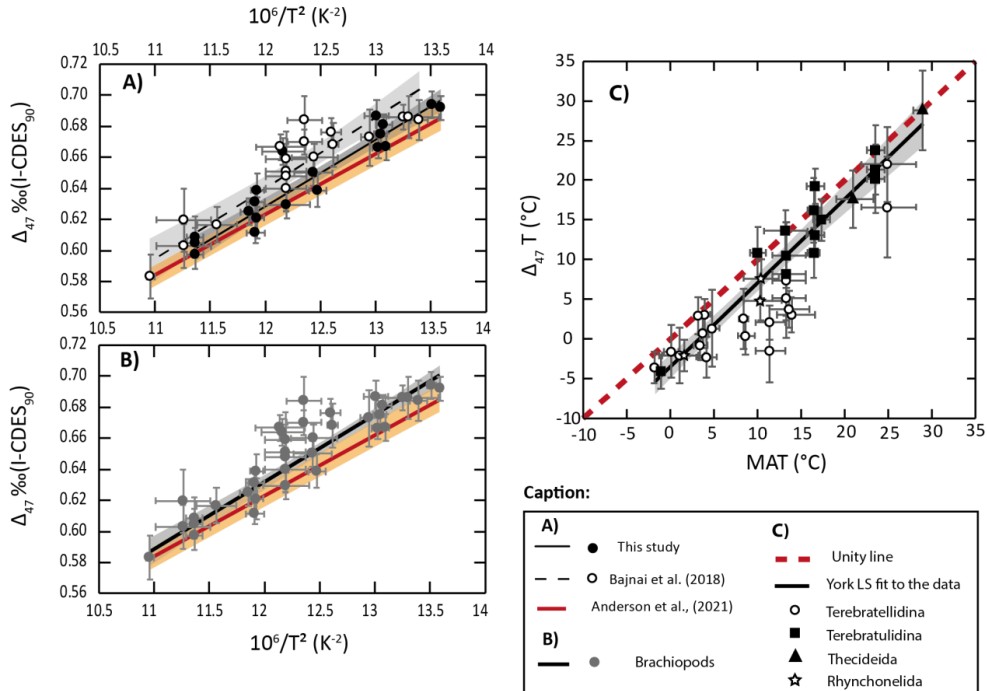

**Figure 4:** Comparison of $\Delta 47$ values from modern brachiopods with state of the art $\Delta 47$ calibration of Anderson et al., (2021). Envelope and error bars at $2\sigma$. A. Comparison of York least square regressions (York et al., 2004) derived from two modern brachiopod datasets (Bajnai et al., 2018; this study) with the equation of Anderson et al., (2021). B. Comparison of York least square regressionderived from the combined modern brachiopod dataset with the equation of Anderson et al., (2021). C) Comparison of modern $\Delta 47$ temperatures calculated using the equation of Anderson et al., (2021) with environmental temperatures.

### 4.1.3 Mg/Ca thermometer for brachiopod shells

Our Mg/Ca data obtained from the inner layers of brachiopod shells do not show any significant correlation with growth temperature (p=0.94; Figure 1C, Table 3). Brand et al. (2013, 2019) proposed that there is a strong correlation between the Mg/Ca molar ratio of brachiopod shell secondary layer and the growth temperature. However, they acknowledge that some species do not follow that trend they illustrated as the GBMgL. They suggest that such deviations resulted from either lower growth rate (Brand et al., 2013) or higher ones (Brand et al., 2019) of the considered species relative to other brachiopods. This issue alone illustrates how from the largest dataset of modern brachiopods available so far (i.e. Brand's database), one cannot firmly determine whether or not Mg/Ca correlates with seawater temperatures. We emphasize that our dataset lack brachiopod shells samples from equatorial surface marine waters (27°C <T< 32°C), which record the highest MgCO$_3$ molar content in Brand's database (Brand et al., 2013, 2019). Interestingly, the highest MgCO$_3$ contents reported by these authors were measured on brachiopods belonging to the order Thecideida. This taxonomic singularity is of prime importance



because: 1) Thecideide brachiopod shells are essentially made of primary shell fabric, secondary shell fabric (fibrous layer) being restricted to small areas of the shell such as the teeth and the inner sockets (Williams, 1968, 1973); 2) the primary shell layer has higher Mg/Ca ratios than the fibrous secondary layer in Rynchonellid and Terebratullid brachiopods (Pérez-Huerta et al., 2008; Gaspard et al., 2018; Rollion-Bard et al., 2019); 3) Ullmann et al., (2017) assigned the high Mg/Ca ratios of Thecideida brachiopod shells to the presence of high-Mg calcite. The exclusion of Thecideida data from the data base published by Brand et al. (2013), produces a substantially weaker ($R^2$=0.23) yet significant (p< 0.01) correlation between shell Mg/Ca and temperature. In addition, magnesium has been shown to be also more concentrated in growth bands (Gaspard et al., 2018; Müller et al., 2022) than in other parts of the secondary shell layer in articulated brachiopods, which could explain the high Mg/Ca values observed from 'slow' growth and long lifespan brachiopods (Brand et al., 2019), especially when considering samples obtained from a large sampling area covering multiple growth lines. Given our new data and all these considerations, the brachiopod Mg/Ca should be regarded as an unreliable proxy for annual or seasonal temperature reconstructions, at least in the -2 to 25°C range (Figure 1, Brand et al., 2013, 2019).

An alternative or complementary way to explain the scattering of temperature-Mg/Ca paired data may rise in the variability of marine Mg/Ca. All the elements considered in this study, (Mg, Sr, Li, Na), have much larger residence times (10 Ma, 3.5 Ma and 1.8 Ma, respectively) than water molecules in the oceans, meaning that at first-order they are uniformly distributed in the main bodies of the oceans (Lécuyer, 2016). A recent study, however, has shown sizable variations in element/Ca may occur in the vicinity of continental margins, under the influence of oceanic currents, at the interface with groundwaters or the atmosphere, or within water masses isolated from the global thermohaline circulation (e.g. Black Sea, Mediterranean Sea, Red Sea) (Lebrato et al., 2020). For example, despite the large residence time of Sr, nonhomogeneous marine strontium isotopic ratios ($^{87}Sr/^{86}Sr$) have been measured in coastal environments (Southern Okinawa Trough, South China Sea and Kao-ping Canyon) as the result of seawater mixing with groundwaters (Huang et al., 2011; El Meknassi et al., 2020). Given the poor constrain on local coastal water chemistry in the deep time, brachiopod samples from offshore environments should be privileged for trace-element-based paleotemperatures reconstructions, as it was suggested for marine biogenic apatite by (Balter and Lécuyer, 2010).

### 4.1.4 Potential of other trace element/Ca proxy

Our new, brachiopod Sr/Ca data show a significant weak negative correlation with ambient temperature ($p_{slope}$ < 0.001; $R^2$ < 0.40). This correlation is supported by a weak positive correlation between Sr/Ca and $\Delta^{18}O_{c-w}$ of the inner layers ($p_{slope}$ < 0.01; $R^2$ = 0.31) as we also expect $\Delta^{18}O$ to decrease with increasing temperatures (i.e section 4.1). However, it contradicts the pioneering findings of Lowenstam, (1961), which suggested a negative correlation between brachiopod shell Sr/Ca and $\Delta^{18}O_{c-w}$. The brachiopod database of Brand et al., (2013) shows no significant correlation between Sr/Ca and temperatures ($p_{slope}$ = 0.20). Given the diverging conclusions from one dataset to the other, Sr/Ca values of brachiopod shells do not appear to be controlled by growing temperatures, at the first order.

Li/Ca in brachiopod shells is significantly correlated to temperature (Figure 1E; Table 3), a covariation initially reported by Delaney et al. (1989) and confirmed by more recent studies (Dellinger et al., 2018; Rollion-



Bard et al., 2019; Washington et al., 2020). Most of the data presented here follow the Li/Ca-temperature trend
       derived from literature data (Washington et al., 2020), with a few samples (n=9), exclusively collected from the
       inner layer, showing significantly lower Li/Ca. Those low Li/Ca values lie below the Li/Ca-temperature
       relationship derived from inorganic calcite (Marriott et al., 2004). Similarly low Li/Ca values were reported from
       brachiopod shells by previous studies (Delaney et al., 1989; Washington et al., 2020). Interestingly, most of those

low Li/Ca values belong to samples from the genus *Tichosina* (Delaney et al., 1989; Washington et al., 2020, Table
       2), so one could consider some peculiar species-dependent trace element partitioning driven by metabolic or kinetic
       processes. Caution is warranted, however, because very low Li/Ca values are also apparent in the genus
       *Stenosarina* Cooper 1977, as well as one specimen of *Aerothyris kerguelenensis* among the 9 specimens studied
       here.

Other environmental parameters may control the element/Ca incorporation into calcite. Sr/Ca, Li/Ca and Na/Ca
       show strong significant negative correlations with the local salinity ($0.37 < R^2 < 0.76$; p-slope <0.01) with similar
       scattering as for the temperature relationships. Mg/Ca again show no significant correlation with this parameter
       (p-slope=0.7). Excluding Mg, element/Ca ratios plotted against living depth revealed two different trends below
       ~100m depth; a group of "low" element/Ca (0.4 < Sr/Ca < 0.7 mmol/mol; 0 < Li/Ca < 10 µmol/mol; 1 < Na/Ca <

4 mmol/mol) and a group of "high" element/Ca brachiopods (0.9 < Sr/Ca < 1.2 mmol/mol; 35 < Li/Ca < 60
       µmol/mol; 8 < Na/Ca < 12 mmol/mol). Such important differences cannot be explained by variations in water
       element/Ca ratios, especially at this depth in the water column (Lebrato et al., 2020). This strong dichotomy, may
       result either from other environmental parameters not considered in this study (e.g., carbonate chemistry), or from
       a strong taxonomic control. Indeed, almost all specimens belonging to the "low" element/Ca group lived in tropical

areas (Guadeloupe, New Caledonia), while all brachiopods belonging to the "high" element/Ca lived at higher
       latitudes (New Zealand, Crozet archipelago, Antarctica). Specimens from New Zealand display the largest
       variability in element/Ca ratios, probably partly because all samples are from coastal environments, where water
       element/Ca are expected to vary the most (Lebrato et al., 2020). All but one of the brachiopods studied here belong
       to the order Terebratulida which regrouped the suborders Terebratulidina and Terebratellidina. All but one

Terebratulidine brachiopod shells are associated with "low" element/Ca ratios, while all but one Terebratellidine
       brachiopod shells are associated with "high" element/Ca ratios; in line with trends in Sr/Ca ratios reported by
       Ullmann et al., (2017). Whether there is an environmental or a taxonomic control on trace element incorporation
       in articulated brachiopods remains unclear but data suggest there may be both. Higher element/Ca ratios were
       measured from *Terebratella sanguinea* (Leach, 1814) (Terebratellidina) than from *Liothyrella neozelanica*

Thomson 1918 (Terebratulidina), while living in the same environment. Similarly, the Antarctic brachiopod
       *Liothyrella uva* (Broderip, 1833) shows the highest Sr/Ca, Na/Ca and Li/Ca ratios, while belonging to the
       Terebratulidine. It is worthy to note that in the dataset presented here, tropical samples are dominated by
       Terebratulidine brachiopods, while high-latitude samples are dominated by Terebratellidine brachiopods, which
       preclude confident testing of both hypotheses from this dataset. In the fossil record, differences in Mg/Ca and

Sr/Ca ratios between brachiopod genera have been associated with different shell microstructures (Ullmann et al.,
       2020).

       Our new data do not support the use of the trace element/Ca ratios studied here as paleothermometers from a
       dataset of widely distributed and taxonomically diverse brachiopods. In other words, temperature is not the prime
       parameter that can describe trace element incorporation into articulated brachiopods from various environments



and belonging to various taxa, suggesting the major influence of other environmental or taxonomic parameters. This conclusion does not exclude the possibility of a strong temperature control on trace element incorporation at the species level as proposed by Butler et al., (2015), and observed in other calcifying organisms such as bivalves (Freitas et al., 2006) and foraminifera (Anand et al., 2003).

**4.2 Out of equilibrium precipitation of brachiopod shell calcite with seawater.**

The oxygen isotope fractionation equation presented here differs from other published equations including that of Kim and O'Neil (1997). However, it is worthy to note that most brachiopod shell data (Brand et al., 2019, this study) plot in-between thermodynamic equilibrium isotopic fractionation (Daëron et al., 2019) and the equation of Kim and O'Neil (1997) (Figure 3). Deviation of brachiopod shell calcite $\delta^{18}O$ from thermodynamic equilibrium

($\Delta^{18}O_{eq} = \delta^{18}O_{brach.calcite} - \delta^{18}O_{equilibrium}$) is obvious when we compare our data set to the equation of Daëron et al. (2019) in the clumped isotope-oxygen isotope fractionation space (Figure 5A) with most of our dataset showing higher $\Delta_{47}$ and lower $\delta^{18}O$ than predicted (Figure 5B), a feature previously highlighted by Bajnai et al., (2018). As a result, brachiopod calcite shows apparent $\Delta_{47}$ and $\delta^{18}O$ temperatures respectively lower and higher than those predicted by the thermodynamic equilibrium. This deviation from equilibrium values has a $\Delta_{47}$-$\Delta^{18}O_{eq}$ slope of

$0.019 \pm 0.005$ in our new brachiopod dataset, indistinguishable from the slope $0.017 \pm 0.003$ reported by Bajnai et al (2018) using $\delta^{18}O_{sw}$ from the $\delta^{18}O_{sw}$ database (LeGrande and Schmidt, 2006).

Brachiopod shell $\Delta^{18}O$ data (Brand et al., 2003, 2013, 2019; this study) also highlight a large geochemical variability among brachiopods sampled from similar marine environment (T,S), with a $\Delta^{18}O$ range often higher than 1‰ (Figure 3, Brand et al., 2019). Isotopic fractionation equations (Brand et al., 2013, 2019; this study)

usually rely on $\Delta^{18}O_{c-w}$ values plotted against mean annual temperatures. Thus, it would be tempting to attribute such scattering of data to seasonal variations, in both temperature and $\delta^{18}O_{sw}$, but only in the case of specimens that lived above or in the upper part of the thermocline. Such isotopic scattering is indeed expected if we consider datasets based on multiple sampling along the growth axis of an individual, such as in the case of Brand et al. (2013; 2019). However, seasonal scattering in data resulting from intra individual geochemical variability is

unlikely for our measurements performed on a large sampling area that should correspond to multiple months or years of growth.

For example, we note a great isotopic variability in both $\delta^{18}O$ and $\Delta_{47}$ among the 9 specimens of *Aerothyris kerguelenensis* from Crozet islands even though the specimens were recovered from depths in the 105 to 355 m range with a narrow range of temperature (3-5°C) and salinity (33.8-34.2). These 9 samples, all collected from the

shell inner layers from different individuals, records a $\Delta^{18}O$ variability of 2‰, equivalent to a temperature range of 8 to 10°C if interpreted as such, and $\Delta_{47}$ variability of 0.02 ‰ equivalent to a temperature range of about 5°C. This subsample shows a significant $\Delta_{47}$-$\Delta^{18}O_{eq}$ slope of $0.027 \pm 0.007$ (2σ; $p_{slope} < 0.01$ ; R²=0.95, n=5), as well as a strong positive correlation between $\Delta^{18}O_{c-w}$ and $\delta^{13}C$ ($p_{slope} < 0.01$ ; R²=0.69, n=9). These covariations most likely result from kinetic effects as defined by McConnaughey (1989) here recorded at the intra-specific level.






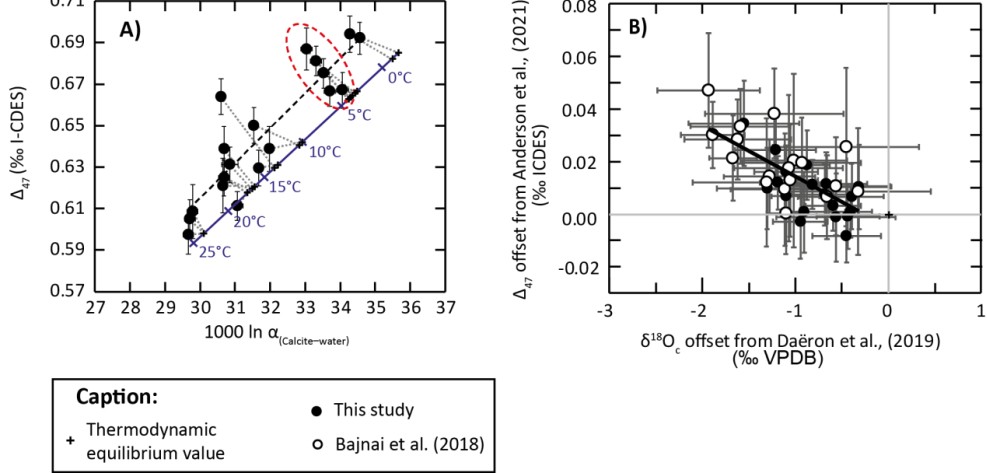

**Caption:**

\+ Thermodynamic equilibrium value  ● This study  ○ Bajnai et al. (2018)

**Figure 5:** Non equilibrium fractionation and kinetic effects in brachiopod shell calcite as evidenced by clumped isotopes and oxygen isotopes. A) Plot of brachiopod inner shell layers in the $\Delta_{47}$ - 1000 ln $\alpha_{(calcite–water)}$ space compared with the equilibrium line (blue line) defined by very slow growing abiogenic calcite (Daëron et al., 2019; Anderson et al., 2021). Red highlight, results from *A. kerguelenensis* (Crozet Islands) and their expected geochemical values on the equilibrium line. Note the negative correlation between $\Delta_{47}$ and 1000 ln $\alpha_{(calcite–water)}$ evidenced in this subset. B) $\delta^{18}O_c$ and $\Delta_{47}$ offset from their expected equilibrium values calculated from environmental parameters using the equations of Daëron et al. (2019) and Anderson et al. (2021), respectively. The trends in these offsets within our new dataset, is similar to what has been reported from other brachiopod samples by Bajnai et al. (2018).

Our new data replicate the observations made by Bajnai et al. (2018) concerning kinetic fractionation in brachiopod shells. In consequence, we follow their conclusions and consider that different growth rates of the specimens are responsible for the second-order isotopic variability and the deviation from stable isotope and clumped isotope equilibrium observed in brachiopod datasets, both at an inter-specific and intra-specific level.

Trace element incorporation into calcite can also be affected by kinetic effects (Lorens, 1981; Busenberg and Plummer, 1985; Tesoriero and Pankow, 1996; Gabitov et al., 2011, 2014). Consequently, if kinetic effects have a strong influence on element incorporation into brachiopod calcite, we should also observe strong negative correlations of Sr/Ca, Na/Ca and Li/Ca ratios with $\Delta^{18}O_{eq}$. We observe such significant correlations in the new dataset, in both the inner and outer layers' subsets ($0.40 < R^2 < 0.55$, p-slope $< 0.01$), with a significant slope change between the two subsets. The variation in element/Ca related to a similar variation in $\Delta^{18}O_{eq}$ as suggested by linear regression models, is about twice as high in the inner layers' subset than in the outer layers' subset. However, while looking at *Aerothyris kerguelenensis,* which showed the most important kinetic effects according to stable isotope data, there is no significant correlation of element/Ca ratios with $\Delta^{18}O_{eq}$. Element/Ca ratios appear to be independent from isotope kinetic effects at the species level, despite strong correlations of element/Ca ratios and $\Delta^{18}O_{eq}$ in the multi-specific dataset.





The deviation from equilibrium isotopic fractionation observed in the new dataset seems to be linked to either taxonomic or environmental factors. As observed when comparing the new data to the equation of Daëron et al. (2019) (Figure 3), deviations from equilibrium oxygen isotope fractionation are greater at lower temperatures, and more generally we observe more deviation from the equilibrium in high latitude environments than in tropical environments. Around Crozet islands, the deeper the brachiopod lived the less deviation from equilibrium is recorded, both in oxygen isotopes and clumped isotopes. While we observe generally more deviation from isotopic equilibrium in Terebratellidina than in Terebratulidina in the whole dataset, the difference is not as striking as documented for trace elements. Given the highly geographically biased distribution of the two suborders in the new dataset, this trend may be coincidental. However, several examples of Terebratellidine and Terebratulidine living in the same environment show similar pattern. In Doubtful Sound (New Zealand) *Terebratella sanguinea* (Terebratellidina) deviates more from equilibrium isotopic composition than *Liothyrella neozelanica* (Terebratulidina) (This study, Bajnai et al., 2018). In Friday Harbor (Washington, USA), *Terebratalia transversa* (Sowerby,1846) (Terebratellidina) records lower and more variable $\delta^{18}O_c$ than *Terabratulina sp.* (Terebratulidina) (Ullmann et al., 2017). These authors also reported high isotopic variability associated with strong correlation of $\delta^{18}O$ with $\delta^{13}C$ in Terebratellidine but not among Terebratulidine. It would seem that brachiopod shells from the suborder Terebratellidina record more important kinetic effects (both from stable isotopes and elemental contents) than brachiopod shells from the suborder Terebratulidina.

Kinetic effects being strongly linked to shell growth rates (McConnaughey, 1989; Bajnai et al., 2018), both taxonomic and environmental parameters should control the extent of the kinetic effects. At first order, one can expect species growing larger shells to record more important kinetic effects. Moreover, shell growth rates in calcifying organisms increase when environmental conditions are close to optimal living range and decrease when the conditions strongly derive from that optimum, a pattern being dependant on taxon-related preferences (Schöne, 2008).

### 4.3. Consequences for the application of $\delta^{18}O$ and $\Delta_{47}$ paleothermometry to brachiopod fossils.

Brachiopod shells display disequilibrium isotopic fractionation in both clumped isotopes and stable isotopes that generates both a general offset in apparent environmental parameters (T, $\delta^{18}O_{sw}$) relative to measured ones, and an increase in the apparent variability of those parameters within a single environment. This is of concern for the accuracy and precision of paleotemperature and $\delta^{18}O_{sw}$ estimates derived from the fossil record.

In this section we illustrate this issue with the study of four fossil brachiopods from the "collections de Géologie de l'Université Claude Bernard Lyon 1", measured alongside the modern brachiopods, and using the same sampling protocol. The four studied specimens were recovered from the Lower Jurassic of Normandy (W-France), in various locations around the cities of Caen and Bayeux. They span the Pliensbachian to Toarcian stages and are well dated with ammonite biostratigraphy. Trace elements measured on those samples revealed Mn and Fe concentrations lower than 150 ppm, and 300 ppm, respectively. Those low values support the good preservation of the fossil material (Brand and Veizer, 1980; Brand et al., 2003; Voigt et al., 2003). In addition, the Lower Jurassic successions of Normandy lies unconformably on the Variscan basement of the Armorican massif and are hence very condensed (~10 m for the whole Pliensbachian-Toarcian) (Dommergues et al., 2008; Weis et al., 2018)





due to the low rates of subsidence in the area since the Triassic (Brunet and Le Pichon, 1982). Hence alteration
of the $\Delta_{47}$ during burial appears unlikely (Henkes et al., 2014; Stolper and Eiler, 2015; Hemingway and Henkes,
2021). Geochemical results and their paleoenvironmental interpretation are displayed in Table 4. To correct from
the geochemical offset recorded in brachiopod shells relative to equilibrium precipitation of calcite, we suggest to
use brachiopod-based calibration for both $\Delta_{47}$ temperatures and $^{18}$O fractionation. Here we applied the equations

derived in this study to this set of fossil brachiopods. $\Delta_{47}$ values recorded in our fossil samples suggest temperatures
of $26 \pm 5$ °C throughout the Pliensbachian and up to $31.8 \pm 5.2$ °C in the late Toarcian. While the $\delta^{18}$O values have
very-good analytical precision ($\pm 0.03$ ‰), they rely on the analysis of only one specimen. Before interpreting the
$\delta^{18}$O data from a single brachiopod specimen, we suggest to use a larger uncertainty of $\pm 1$‰ to account for the
natural variability of $\delta^{18}$O values within a brachiopod population (this study, Brand et al., 2019). Accordingly,

$\delta^{18}O_{sw}$ estimates from single brachiopod specimen are associated with uncertainties about twice as large as if only
analytical uncertainties were considered. Here, Pliensbachian to Toarcian $\delta^{18}O_{sw}$ values estimated for W France
range from $0.3 \pm 1.7$ to $1.1 \pm 2.0$ ‰ VSMOW. The uncertainties associated with $^{18}O_{sw}$ estimates determine the
range of temporal or spatial variations in $\delta^{18}O_{sw}$ that could be confidently identified from brachiopod fossils. Given
the very large uncertainties we thus suggest to analyse multiple well-preserved specimens within a single bed

(Ullmann et al., 2020), in order to characterise the geochemical distribution of the studied population, and use
central values with their associated uncertainties, for paleoenvironmental interpretations (at least for $\delta^{18}O_c$). In this
case, the classically-used positive correlation between $\delta^{18}O_c$ and $\delta^{13}C$ (McConnaughey, 1989) and a negative
correlation between $\delta^{18}O_c$ and $\Delta_{47}$ (Bajnai et al., 2018, this study) would also provide solid evidence for kinetic
effects. Finally, kinetic biases may be corrected through the use of dual clumped isotope thermometry (Bajnai et

al., 2020). However, given their instrumental and temporal costs, such clumped isotope methods will require
further analytical development before they can be routinely used to detect and correct such biases in the large
number of samples needed for paleoclimate studies.



 

**Table 4:** Fossil brachiopods' geochemical results and paleoenvironmental interpretations with $2\sigma$ analytical and propagated uncertainties. Both brachiopod specific $\Delta_{47}$ calibration and the Anderson et al. (2021) are compared on these samples. $\delta^{18}O_{sw}$ are calculated from $\Delta_{47}$ T and $\delta^{18}O_c$ values using the brachiopod specific equation derived in this study. An additional $\sim 1$‰ may be added to $\delta^{18}O$ uncertainties and then propagated, to consider the natural dispersal of brachiopod $\delta^{18}O$ values observed within modern and fossil brachiopod populations, which is one to two orders of magnitude larger than analytical uncertainties.

| Sample | FSL 707721 | FSL 707727 | FSL 707729 | FSL 707752 |
|---|---|---|---|---|
| Species | *Plarorhynchia thalla* | *Tertrarhynchia dumbletonensis* | *Quadratirhynchia quadrata* | *Homeorhynchia cynocephala* |
| Location | Subles | Feuguerolles | Tilly-sur-Seulles | Subles |
| Stage | Pliensbachian | Pliensbachian | Pliensbachian | Toarcian |
| Ammonite Biozone | Jamesoni | Margaritatus | Spinatum | Aalensis |
| ***Geochemical results*** | | | | |
| $\delta^{13}C$ (‰ VPDB) | 1.16 ± 0.02 | 2.40 ± 0.02 | 1.49 ± 0.02 | 1.33 ± 0.02 |
| $\delta^{18}O$ (‰ VPDB) | − 0.55 ± 0.03 | − 0.24 ± 0.03 | − 0.15 ± 0.03 | − 0.93 ± 0.03 |
| $\Delta_{47}$ (‰ ICDES) | 0.600 ± 0.011 | 0.602 ± 0.016 | 0.600 ± 0.016 | 0.583 ± 0.016 |
| Mn (ppm) | 26 | 42 | 121 | 50 |
| Fe (ppm) | 39 | 88 | 282 | 198 |
| ***Paleoenvironment interpretation (this study)*** | | | | |
| $\Delta_{47}$ T (°C) | 26.1 ± 3.4 | 25.4 ± 4.9 | 26.2 ± 4.9 | 31.8 ± 5.1 |
| $\delta^{18}O_{sw}$ (‰ VSMOW) | 0.92 ± 0.68 | 1.09 ± 0.97 | 1.33 ± 0.98 | 1.63 ± 1.03 |
| ***Paleoenvironment interpretation (Anderson et al., 2021)*** | | | | |
| $\Delta_{47}$ T (°C) | 22.9 ± 3.5 | 21.1 ± 5.1 | 22.9 ± 5.1 | 28.9 ± 5.5 |
| $\delta^{18}O_{sw}$ (‰ VSMOW) | 0.29 ± 0.71 | 0.46 ± 1.01 | 0.70 ± 1.02 | 1.05 ± 1.08 |

### 5. Conclusion

Our multi-proxy ($\delta^{18}O$ and $\delta^{13}C$, $\Delta_{47}$, Mg/Ca, Sr/Ca, Na/Ca and Li/Ca) geochemical analyses of 37 modern brachiopods from different locations, spanning living temperatures from -1.9°C to 25.5°C confirm that the brachiopod shell inner layers should be preferred to estimate water properties. There is a strong correlation between the living temperature and the $\Delta^{18}O$ fractionation between the brachiopod shell inner layers and water that allowed us to establish a new $\delta^{18}O$ paleotemperature equation for brachiopod shell calcite. Our new trace element data show no significant correlation between growing temperature and Mg/Ca ratios measured from the inner shell



layers indicating that brachiopod shell Mg/Ca is not a suitable paleotemperature proxy. Sr/Ca, Li/Ca and Na/Ca of the brachiopod shell inner layers show a significant negative correlation with growing temperatures. However, partitioning of those elements between the low-Mg calcite brachiopod shell and seawater appears to show strong taxonomic patterns. Clumped isotope temperatures, derived from the calibration of (Anderson et al., 2021) are generally underestimated by ~3°C relative to the environmental temperature. In consequence, we suggest to use a

calibration especially dedicated to the brachiopods for reconstruction seawater temperatures. In addition, our new $\Delta_{47}$ data confirm previous evidence for non-equilibrium isotopic fractionation interpreted as significant kinetic effects recorded in the brachiopod shell inner layers. In general, the precipitation of brachiopod shells taking place out of isotopic equilibrium leads to 1) an under-estimation of living temperature from $\Delta_{47}$, 2) an over-estimation of living temperature from oxygen isotopes and 3) an underestimation of living $\delta^{18}O_{sw}$ when combining the two

proxies, relative to what would be expected from equilibrium precipitation. Locally, the kinetic effects produce geochemical apparent temperatures with high variability relative to the local temperature range, for both the oxygen isotope and the clumped isotope proxies. We therefore advise to consider this natural variability in the interpretation of paleoenvironmental parameters from brachiopod shell $\delta^{18}O$ and $\Delta_{47}$ data.

**Author contribution**

TL and CL designed the study in close collaboration with GS and DG, DG obtained the samples from various contributors, TL performed shell sampling in the lab, AVL and TL performed the stable isotope analysis and data treatment, TL and FAG performed the elemental analysis and data treatment, TL and MD performed the clumped isotope analysis and data treatment, TL performed the statistical analysis of the data and wrote the first draft.


**Acknowledgments**

This study was founded by the ANR-OXYMORE (grant no. ANR-18-CE31-0020)

The authors gratefully thank the leaders and teams of the N/O of the following expeditions: the Walda cruise in S.E. Atlantic Ocean (Dr. M. Segonzac, IFREMER, Brest); Sampling from Norway coasts (courtesy of the

Trondheim Biological station); MD 30 "BIOMASS" around Crozet Islands (Arnaud, Marseille, France and Prof N. Ameziane, MNHN, who provided the material); Musorstom 4, SMIB 3 and 6 and Norfolk 1 around New Caledonia, Loyauté - Lifou Islands (Prof. B. Laurin, Univ. Bourgogne, Dijon and Richer de Forges); Karubenthos 2, around Guadeloupe: French Caribbean Island (Prof. Ph. Bouchet with Dr. P. Lozouet, Dr. L. Corbari and Ph. Maestrati and the help of J. Mainguy, MNHN, Paris, France) with a supplementary sampling (Prof. D. Lamy,

Univ. Antilles-Guyane); supplementary material from New-Zealand (Prof. Lee, D., Univ Otago, Marine Biology Lab.; Prof. D.I. MacKinnon and Dr. A. Aldridge (Christchurch Univ.); CEAMARC off French Antarctic area (Prof. N. Ameziane and Dr. M. Eléaume, MNHN, Paris, France); sampling from Palmer Peninsula, Antarctic (courtesy of Th. Desvignes, USA).

We thank Emmanuel Robert for granting access to fossil specimens from the "collections de Géologie de

l'Université Claude Bernard Lyon 1".



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
