# Peer review of "Multi-proxy assessment of brachiopod shell calcite as a potential archive of seawater temperature and oxygen isotope composition"

_EGUsphere, 2022_

## Author Comment (AC1)

**Authors response to referee comment 1**

Dear editor, dear referee, dear EGU sphere,

First we would like to thank the referee for the time invested reviewing our paper, and the constructive comments and suggestions that were made.
We take this opportunity to answer the most important questioning and suggestions in detail. We took notes of other minor comments to correct and improve the manuscript in its revised form.

We start answering the main concern raised by referee 1, concerning the taxon-specific clumped isotope temperature equation which was the object of several comments.

*My only major disagreement with the author's assessment is the need for a taxon-specific calibration for clumped isotope-temperature reconstructions in brachiopods. In my opinion, the authors arrive at this conclusion a bit too hastily, only comparing their findings to one pre-existing clumped isotope-temperature calibration (by Anderson et al.,2021), even though previous studies have noted that this equation tends to underestimate temperatures (see detailed comments below). This conclusion of a brachiopod-specific paleotemperature equation is, in my opinion, also in disagreement with the conclusions of disequilibrium fractionation in brachiopods put forward by papers by David Bajnai (which are cited and discussed in the manuscript). I do not agree with the approach of circumventing this disequilibrium issue by inventing a new empirical temperature equation for using clumped isotopes for temperature reconstructions from brachiopod calcite. I therefore think the authors should revise this conclusion before the paper becomes acceptable.*

*Line 348-351: I suggest the authors also compare the clumped isotope-temperature relationship in brachiopod calcite with the values obtained by applying the Meinicke et al. (2020) calibration, which was recently updated to the I-CDES scale (Meinicke et al.,2021). A recent study by de Winter et al. (2022) demonstrated that the Anderson et al. equation likely induces a cold bias on shallow water carbonates which might explain part of the observed offset in clumped isotope values in this study. I wonder if the brachiopod specific clumped isotope equation proposed by the authors is significantly different from Meinicke et al. if projected on the I-CDES scale. If so, it might not be warranted to propose a new clumped isotope calibration as brachiopods might be calibrated with general calcite calibrations.*

*Line 623-624: The temperature underestimation by ~3 degrees is very similar to the offset found in de Winter et al. (2022; see comment above) and this nice corroboration between multiple datasets is worth mentioning.*

*Line 624-625: As mentioned above, I tend towards disagreeing with this call for a brachiopod-specific clumped isotope calibration, since most of the offset in clumped isotope values may be explained by the Anderson et al. equation underestimating temperatures in general (not just for brachiopods). The authors should consider this explanation before suggesting a taxon-specific calibration is needed.*

Reviewer #1 argues against our conclusion that our modern brachiopods observations are inconsistent with "general" calcite calibrations, proposing that we should compare them to a recently reprocessed foraminifer calibration (Meinicke et al., 2021), which predicts slightly greater calcite Δ47 values for temperatures below ~10 °C.

For reasons outlined below, we would argue that (1) Anderson et al. (2021) is in fact fully consistent with most other I-CDES data sets, and (2) the difference between Meinicke et al. (2021) and other calibrations is due to problematic choices in the assignment of calcification temperatures.

(1) We'd first like to point out that our use of the Anderson et al. (2021), despite being a rather consensual choice, is but one of many that would lead to an identical conclusion. For instance, Fig 3B from Anderson et al. illustrates the quasi-perfect agreement between their regression line and the low-temperature observations of Peral et al. (2018, foraminifera), Meinicke et al. (2020, foraminifera), Jautzy et al. (2020, synthetics), and slow-growing calcites from Laghetto Basso and Devils Hole. Based on this, we find it extremely difficult to claim that these five independent calibration datasets are inconsistent in any statistically significant way. A sixth data set on modern bivalves (Huyghe et al., 2022), as well as new measurements of slow-growing calcites (Fiebig et al., 2021) further strengthen the case that all of these calibration data, when (re)processed to the I-CDES, are fully consistent (Figure 1).

(2) At face value, this would seem to contradict the assertion that the I-CDES regression equation from Meinicke et al. (2021) yields substantially greater Δ47 values for cold temperatures. Yet, as noted in the original study by Meinicke et al. (2020) themselves, there is no disagreement between the foraminifer datasets from LSCE (Peral et al., 2018) and from Bergen (Meinicke et al., 2020 and other papers cited therein). The difference between the corresponding (reprocessed) I-CDES calibrations (Peral et al., 2022; Meinicke et al., 2021) is thus due to different approaches of assigning Δ47-independent calcification temperatures to planktic foraminifera (as discussed in detail by Meinicke et al., 2020). This in itself is not a problem: each of the two approaches appear reasonable a priori. However, we now have many precise Δ47 measurements for samples precipitated between -2 and +8 °C, including benthic foraminifera (Peral et al., 2018), Antarctic scallops (Huyghe et al., 2022), lacustrine carbonates from perenially ice-covered lakes (Anderson et al., 2021), and very slow-growing calcite from Laghetto Basso (Anderson et al., 2021; Fiebig et al., 2021), all of which have very narrow temperature constraints, and which agree extremely well with the original temperature assignments of Peral et al. (2018). Based on all of this evidence, it is very likely that the apparent discrepancy between Meinicke et al. (2021) and the other calibrations primarily reflects problematic choices in the assignment of calcification temperatures.

[Figure]

Various calibration data sets, reprocessed into the I-CDES scale using the D47crunch library. Error bars are 95 % confidence limits.

Figure 1: Plot comparison of the new brachiopod clumped isotope data with published datasets from foraminifera, calcite bivalves, slow growing calcite and synthetic carbonates, in the I-CDES reference frame and their associated regression lines. This illustrate the good agreement between brachiopods and previous datasets in the temperature range 10 to 25°C, at one exception. In the low temperature range (−2 - 7°C) however the brachiopod dataset shows significant differences.

Reviewer #1 also argues against the need for a taxon-specific equation. In this regard, while we have independently confirmed the conclusions of Bajnai et al. (2018) regarding the deviation from previous calibration, we agree that establishing a taxon-specific equation may not be the adequate way to correct for this deviation. Indeed, as discussed and illustrated in Figure 4 C of the manuscript, while we observe more deviation from the equation of Anderson et al., 2021 in the low temperature range (<10°C), this could also well be explained by biologic parameters, such as different growth rates between different taxa, as suggested by Bajnai et al. (2018). And as further discussed, temperature is not the only parameters controlling brachiopod shell growth rates. In line with the suggestion of reviewer#1, we will thus retract on suggesting the use a taxon-specific equation, but rather focus on the potential explanations for this discrepancy, and better tie this result the rest of the discussion (section 4.2).

To give a brief answer to this major comment by reviewer #1. For various reasons stated above we further assure our conclusion that the empirical Δ47-temperature relationship observed in brachiopod shells shows significant differences to previous relationships established on carbonates from different origins (foraminifera, calcite bivalves, synthetic) and best illustrated by the equation of Anderson et al. (2021). However, this conclusion is not sufficient to propose a taxon-specific equation to be applied to brachiopods shell, but rather highlight other controls on Δ47 values of carbonates than temperatures, that should be further investigated.

*In the Introduction (lines 49-89), several potential temperature proxies in brachiopod calcite are introduced one by one. While these introductions are important and wellwritten, the individual paragraphs are quite detached from each other and disrupt the flow of the*

*manuscript somewhat. I suggest the authors either tie the paragraph a bit better into the rest of the manuscript or place them in a separate "Background" paragraph.*

We will rework the introduction to better tie the paragraph into the rest of the MS.

*Lines 124-129: Perhaps the authors can provide a citation or reason for why they used this pre-treatment method. There is some literature suggesting that pre-treatment with oxidizing agents might influence the (clumped) isotope or trace element composition of the carbonate. Personally, I am of the opinion that excess pre-treatment with such substances should be avoided in these types of studies. However, if the authors have convincing evidence (either by their own research or from the literature) that this treatment is warranted in this case, I am happy to support it.*

The literature appears divided on the subject (Schöne et al., 2017; Key et al., 2020; for the most recent litterature). We have chosen to perform an oxidizing pre-treatment as most of the samples still had the animal within their shell. Most of the organic matter was removed manually but an oxidizing pre-treatment was deemed necessary to remove organic matter still attached to the shell before sampling for shell geochemistry. Although, we acknowledge that this could have been avoided for samples without clear evidence for organic material. Among the different pre-treatment method used in previous studies, we adopted a protocol close to the one used by Bajnai et al., (2018) as we preferred NaClO over $H_2O_2$ for logistical reasons. We note that bleaching is associated with little to no carbonate dissolution as opposed to H2O2 (Pingitore et al., 1993; Gaffey and Bronnimann, 1993).

*Section 3.1: It is not clear from this section how the authors dealt with uncertainty on the temperature and salinity/d18Ow value associated with the samples. Since brachiopod samples are large and do not always represent mean annual averages (i.e. due to sampling of less than a full year, see lines 144-146, or due to variations in growth rate over the year) I think the authors should take into account the seasonal cycle in temperature and d18Ow (or salinity) at the sampling locality as uncertainty on their regression. I assume from the text in this section that the authors used a normal linear regression (not including errors on measurements or on the independent variables). To incorporate uncertainty on the independent variable, the authors could use a Deming regression which takes into account measurement error as well as errors on the "known" variable (in this case temperature and d18Ow). Judging from Figure 1 (which I assume shows uncertainties on temperature), these uncertainties are significant.*

The uncertainties displayed in Figure 1 corresponds to the seasonal variation for temperature (highest and lowest monthly averages), and for $\delta^{18}O_w$ the uncertainty combines uncertainty of the salinity- $\delta^{18}O_w$ relationship (LeGrande and Schmidt, 2006) and propagated seasonal variation in salinity, although it is very low for most samples. The regression presented in Figure 1 and Table 3 are simple linear regressions which are primarily used to explore the dataset. While the linear model was proposed in the submitted manuscript for the fractionation equation, we take note of the point made here, and will rather display the results of the York regression (York et al., 2004), which account for uncertainties on the dataset, although in this particular case, it is not significantly different than the linear model. Note that this model is introduced and discussed later on in the submitted manuscript. We will also add some details as to how uncertainties were considered.

**Minor comments**

*Line 262-263: "At temperate and polar temperatures (20 to 0°C) our equation has a steeper slope than that of Brand et al. (2019)" The authors need to explain this in more detail and/or refer to a figure where the reader can spot this effect. The authors refer to figure 2A later, but it is not clear which line in this figure represents the equation by Brand. I do not understand how the slope can be different with temperature if a linear equation (with a constant slope) is compared, but I might misunderstand what the authors are trying to say. Later on, the authors mention that the Brand et al. equation is non-linear, but I still have trouble following the description of the comparison in this section.*

Another referee also stated that this part was unclear. We will address this issue in the revised manuscript and provide further and clearer discussion regarding the difference in "slope" of different equations.

*Line 443-466: This clear grouping based on trace element content is an interesting observation. I wonder if the authors considered whether there might be a relationship with growth rate. Does the "high" or the "low" group show significantly faster growth than the other? If so, this could be an explanation for the difference in shell composition, as trace element concentrations in calcifiers often show a correlation with growth rate. In addition, in this section about grouping of specimens based on trace element content, adding a figure showing the differences in concentration would be helpful.*

*Lines 516-531: It seems that some of the discussion of kinetic (growth rate-related) effects in trace element composition could be a useful addition to the section above where the observation of differences in trace element composition between brachiopod groups is discussed (see my comment on lines 443-466).*

This trend may be related to growth rates. Unfortunately, our new dataset comprises only 5 species for which we have constraints on their growth rate, limiting quantitative comparisons.. This hypothesis could be mentioned into the discussion. A figure will be added to illustrate this dichotomy. We must also note that this difference is confined to the inner shell layers. The trends described here between Terebratellidina and Terebratulidina disappear while looking at the outer shell layers.

*Line 493-495: The authors might consider citing the recent study by Garbelli et al. (2022) here who also interpret changes in isotopic composition of (fossil) brachiopod shells as seasonal variability.*

*Garbelli, C., Angiolini, L., Posenato, R., Harper, E. M., Lamare, M. D., Shi, G. R., and Shen, S.: Isotopic time-series (δ13C and δ18O) obtained from the columnar layer of Permian brachiopod shells are a reliable archive of seasonal variations, Palaeogeography, Palaeoclimatology, Palaeoecology, 607, 111264, https://doi.org/10.1016/j.palaeo.2022.111264, 2022.*

We agree that the addition of literature regarding seasonal variability registered in brachiopod shells would be pertinent here. Although very interesting and convincing, the work of Garbelli et al. (2022) may not be the most pertinent reference here, when other papers studying modern brachiopods in monitored environment unambiguously highlight seasonal variability (Yamamoto et al., 2011; Takayanagi et al., 2015).

*Line 539-540: "Given the highly…may be coincidental." I think the authors should explain this line of reasoning in a bit more detail.*

Here we intend to provide a critical look at our dataset as the differences in both trace elements and clumped isotopes may be explained by two kinds of groupings in our dataset: 1)High latitudes vs low latitudes with difference in seasonality of the ecosystem that may induce different growth rate dynamics and 2) taxonomic grouping which may involve differences in shell formation processes including growth rate. However, as the Terebratulidina are mostly associated with low latitudes and Terebratellidina with mid-high latitudes in the dataset, we lack strong arguments to prefer a hypothesis over the other. We will rephrase this part to make it easier to understand.

*Lines 556-610: I think the addition of a fossil case study is nice, but it is not essential for the study. If the authors would like to keep their manuscript more concise, this is a section that could be significantly shortened or removed in my opinion.*

Another referee has done a similar comment. This section will be removed from the revised version.

**References**

Anderson, N. T., Kelson, J. R., Kele, S., Daëron, M., Bonifacie, M., Horita, J., Mackey, T. J., John, C. M., Kluge, T., Petschnig, P., Jost, A. B., Huntington, K. W., Bernasconi, S. M., and Bergmann, K. D.: A Unified Clumped Isotope Thermometer Calibration (0.5–1,100°C) Using Carbonate-Based Standardization, Geophys. Res. Lett., 48, https://doi.org/10.1029/2020GL092069, 2021.

Bajnai, D., Fiebig, J., Tomašových, A., Milner Garcia, S., Rollion-Bard, C., Raddatz, J., Löffler, N., Primo-Ramos, C., and Brand, U.: Assessing kinetic fractionation in brachiopod calcite using clumped isotopes, Sci. Rep., 8, 533, https://doi.org/10.1038/s41598-017-17353-7, 2018.

Fiebig, J., Daëron, M., Bernecker, M., Guo, W., Schneider, G., Boch, R., Bernasconi, S. M., Jautzy, J., and Dietzel, M.: Calibration of the dual clumped isotope thermometer for carbonates, Geochim. Cosmochim. Acta, 312, 235–256, https://doi.org/10.1016/j.gca.2021.07.012, 2021.

Gaffey, S. J. and Bronnimann, C. E.: Effects of bleaching on organic and mineral phases in biogenic carbonates, J. Sediment. Res., 63, 752–754, https://doi.org/10.1306/D4267BE0-2B26-11D7-8648000102C1865D, 1993.

Huyghe, D., Daëron, M., de Rafelis, M., Blamart, D., Sébilo, M., Paulet, Y.-M., and Lartaud, F.: Clumped isotopes in modern marine bivalves, Geochim. Cosmochim. Acta, 316, 41–58, https://doi.org/10.1016/j.gca.2021.09.019, 2022.

Jautzy, J. J., Savard, M. M., Dhillon, R. S., Bernasconi, S. M., and Smirnoff, A.: Clumped isotope temperature calibration for calcite: Bridging theory and experimentation, Geochem. Perspect. Lett., 36–41, https://doi.org/10.7185/geochemlet.2021, 2020.

Key, M. M., Smith, A. M., Phillips, N. J., and Forrester, J. S.: Effect of removal of organic material on stable isotope ratios in skeletal carbonate from taxonomic groups with complex mineralogies, Rapid Commun. Mass Spectrom., 34, https://doi.org/10.1002/rcm.8901, 2020.

LeGrande, A. N. and Schmidt, G. A.: Global gridded data set of the oxygen isotopic composition in seawater, Geophys. Res. Lett., 33, L12604, https://doi.org/10.1029/2006GL026011, 2006.

Meinicke, N., Ho, S. L., Hannisdal, B., Nürnberg, D., Tripati, A., Schiebel, R., and Meckler, A. N.: A robust calibration of the clumped isotopes to temperature relationship for foraminifers, Geochim. Cosmochim. Acta, 270, 160–183, https://doi.org/10.1016/j.gca.2019.11.022, 2020.

Meinicke, N., Reimi, M. A., Ravelo, A. C., and Meckler, A. N.: Coupled Mg/Ca and Clumped Isotope Measurements Indicate Lack of Substantial Mixed Layer Cooling in the Western Pacific Warm Pool During the Last ~5 Million Years, Paleoceanogr. Paleoclimatology, 36, https://doi.org/10.1029/2020PA004115, 2021.

Peral, M., Daëron, M., Blamart, D., Bassinot, F., Dewilde, F., Smialkowski, N., Isguder, G., Bonnin, J., Jorissen, F., Kissel, C., Michel, E., Vázquez Riveiros, N., and Waelbroeck, C.: Updated calibration of the clumped isotope thermometer in planktonic and benthic foraminifera, Geochim. Cosmochim. Acta, 239, 1–16, https://doi.org/10.1016/j.gca.2018.07.016, 2018.

Peral, M., Bassinot, F., Daëron, M., Blamart, D., Bonnin, J., Jorissen, F., Kissel, C., Michel, E., Waelbroeck, C., Rebaubier, H., and Gray, W. R.: On the combination of the planktonic foraminiferal Mg/Ca, clumped ($\Delta_{47}$) and conventional ($\delta^{18}O$) stable isotope paleothermometers in palaeoceanographic studies, Geochim. Cosmochim. Acta, 339, 22–34, https://doi.org/10.1016/j.gca.2022.10.030, 2022.

Pingitore, N. E., Borrego, P. M., and Crawford', G. M.: Dissolution kinetics of $CaCO_3$ in common laboratory solvents, J. Sediment. Petrol., 63, 641–645, 1993.

Schöne, B. R., Schmitt, K., and Maus, M.: Effects of sample pretreatment and external contamination on bivalve shell and Carrara marble $\delta^{18}O$ and $\delta^{13}C$ signatures, Palaeogeogr. Palaeoclimatol. Palaeoecol., 484, 22–32, https://doi.org/10.1016/j.palaeo.2016.10.026, 2017.

Takayanagi, H., Asami, R., Otake, T., Abe, O., Miyajima, T., Kitagawa, H., and Iryu, Y.: Quantitative analysis of intraspecific variations in the carbon and oxygen isotope compositions of the modern cool-temperate brachiopod Terebratulina crossei, Geochim. Cosmochim. Acta, 170, 301–320, https://doi.org/10.1016/j.gca.2015.08.006, 2015.

Yamamoto, K., Asami, R., and Iryu, Y.: Brachiopod taxa and shell portions reliably recording past ocean environments: Toward establishing a robust paleoceanographic proxy: BRACHIOPOD OXYGEN ISOTOPE RECORDS, Geophys. Res. Lett., 38, L13601, https://doi.org/10.1029/2011GL047134, 2011.

---

## Author Comment (AC3)

**Authors response to referee comment 3**

Dear editor, dear referee, dear EGU sphere,

First we would like to thank the referee for the time invested reviewing the paper, and the constructive comments and suggestions that were made.
Here, we answer in details some of the comments. We took notes of the other suggestions regarding language, or small additions.

*Comparison of the new equation of the oxygen isotope fractionation with those of Brand et al. (2019). Both equations are similar in the temperature range between 10-25°C, but differ in the low-temperature field (<10°C). Since the Brand et al brachiopod-data set is by far much better constrained by data points, the authors should provide a more in-depth discussion about causes of the offset. Further, they should strengthen their arguments why it is necessary to introduce the new equation, and why it shall be an improvement.*

We will clarify this part and add elements of comparison between the different fractionation equations.

*The supposed sampling procedure avoids specialized parts of the shell as umbo, edges, muscle scars, primary layers. However, Ullmann et al. (2017) observed additional significant taxon-specific ranges in their intra-specific high-resolution oxygen isotope data. How has this observation an effect for the results of this study? Is intra-specific variability smoothed by the sample size? Which degree of uncertainty introduce specimens of the suborder Terebratellidina to the fractionation equations? Please comment on this.*

We assume that the sampling area, which spans at least two major growth lines, allows a smoothing of the intra-specimen variability within the collected homogenized powder. While major growth lines are commonly interpreted as annual, that is not necessarily the case, which explains the uncertainty on the time represented by the sampling area. This sampling creates mass-averaged geochemical compositions, which is mathematically different from time-averages which we use for environmental conditions, because of varying shell growth rates through time. The more variable the growth rate, the more biased this sampling will be towards periods of high growth rates (Schöne, 2008; Yamamoto et al., 2011; Takayanagi et al., 2015).
Specimens of the suborder Terebratellidina highlight clumped isotope deviations from equilibrium and trace elements compositions of their inner layer significantly different than specimens form the suborder Terebratulidina. However, deviation from thermodynamic equilibrium in oxygen isotope composition are not significantly different between the two groups, within the dataset tested. So we do not expect the fractionation equation to be biased by any of the groups.

*The Jurassic example is not well executed and not the scope of this study. New and very few data points are introduced first time in the discussion. I recommend the removal of this part of the manuscript, since its focus is on modern brachiopod taxa.*

As also suggested by another referee, this part will be removed from the revised version of the manuscript.

**References**

Schöne, B. R.: The curse of physiology—challenges and opportunities in the interpretation of geochemical data from mollusk shells, Geo-Mar. Lett., 28, 269–285, https://doi.org/10.1007/s00367-008-0114-6, 2008.

Takayanagi, H., Asami, R., Otake, T., Abe, O., Miyajima, T., Kitagawa, H., and Iryu, Y.: Quantitative analysis of intraspecific variations in the carbon and oxygen isotope compositions of the modern cool-temperate brachiopod Terebratulina crossei, Geochim. Cosmochim. Acta, 170, 301–320, https://doi.org/10.1016/j.gca.2015.08.006, 2015.

Yamamoto, K., Asami, R., and Iryu, Y.: Brachiopod taxa and shell portions reliably recording past ocean environments: Toward establishing a robust paleoceanographic proxy: BRACHIOPOD OXYGEN ISOTOPE RECORDS, Geophys. Res. Lett., 38, L13601, https://doi.org/10.1029/2011GL047134, 2011.

---

## Author Response (AR1)

**Author's response file.**

All line calls in this file refer to the Author's track change file.
It is presented as follow:
- Referee comments
- Author's response
- Changes to the manuscript

**Comments from Anonymous Referee #1**

In the Introduction (lines 47-107), several potential temperature proxies in brachiopod calcite are introduced one by one. While these introductions are important and well written, the individual paragraphs are quite detached from each other and disrupt the flow of the manuscript somewhat. I suggest the authors either tie the paragraph a bit better into the rest of the manuscript or place them in a separate "Background" paragraph.

We will rework the introduction to better tie the paragraph into the rest of the MS.

lines 47-107: Several minor additions were made to the paragraphs concerning the different potential proxies, with the aim to better tie them together.

Lines 156-161: Perhaps the authors can provide a citation or reason for why they used this pre-treatment method. There is some literature suggesting that pre-treatment with oxidizing agents might influence the (clumped) isotope or trace element composition of the carbonate. Personally, I am of the opinion that excess pre-treatment with such substances should be avoided in these types of studies. However, if the authors have convincing evidence (either by their own research or from the literature) that this treatment is warranted in this case, I am happy to support it.

The literature appears divided on the subject (Schöne et al., 2017; Key et al., 2020; for the most recent literature). We have chosen to perform an oxidizing pre-treatment as most of the samples still had the animal within their shell. Most of the organic matter was removed manually but an oxidizing pre-treatment was deemed necessary to remove organic matter still attached to the shell before sampling for shell geochemistry. Although, we acknowledge that this could have been avoided for samples without clear evidence for organic material. Among the different pre-treatment method used in previous studies, we adopted a protocol close to the one used by Bajnai et al., (2018) as we preferred NaClO over $H_2O_2$ for logistical reasons. We note that bleaching is associated with little to no carbonate dissolution as opposed to $H_2O_2$ (Pingitore et al., 1993; Gaffey and Bronnimann, 1993).

There is no modification associated with this comment.

Section 3.1: It is not clear from this section how the authors dealt with uncertainty on the temperature and salinity/d18Ow value associated with the samples. Since brachiopod samples are large and do not always represent mean annual averages (i.e. due to sampling of less than a full year, see lines 144-146, or due to variations in growth rate over the year) I think the authors should take into account the seasonal cycle in temperature and d18Ow (or salinity) at the sampling locality as uncertainty on their regression. I assume from the text in this section that the authors used a normal linear regression (not including errors on measurements or on the independent variables). To incorporate uncertainty on the independent variable, the authors could use a Deming regression which takes into account measurement error as well as errors on the "known" variable (in this case temperature and d18Ow). Judging from Figure 1 (which I assume shows uncertainties on temperature), these uncertainties are significant.

The uncertainties displayed in Figure 1 corresponds to the seasonal variation for temperature (highest and lowest monthly averages), and for $\delta^{18}O_w$ the uncertainty combines uncertainty of the salinity- $\delta^{18}O_w$ relationship (LeGrande and Schmidt, 2006) and propagated seasonal variation in salinity, although it is very low for most samples. The

regression presented in Figure 1 and Table 3 are simple linear regressions which are primarily used to explore the dataset. While the linear model was proposed in the submitted manuscript for the fractionation equation, we take note of the point made here, and will rather display the results of the York regression (York et al., 2004), which account for uncertainties on the dataset, although in this particular case, it is not significantly different than the linear model. Note that this model is introduced and discussed later on in the submitted manuscript. We will also add some details as to how uncertainties were considered.

Line 255-261: The oxygen isotope fractionation equation displayed here and in the abstract (line 28) corresponds to the result of a York regression (York et al., 2004) which considers the uncertainties associated with each data point in the regression model. We also added precision on the parameters we used to set the uncertainties.

Line 348-351: I suggest the authors also compare the clumped isotope-temperature relationship in brachiopod calcite with the values obtained by applying the Meinicke et al. (2020) calibration, which was recently updated to the I-CDES scale (Meinicke et al., 2021). A recent study by de Winter et al. (2022) demonstrated that the Anderson et al. equation likely induces a cold bias on shallow water carbonates which might explain part of the observed offset in clumped isotope values in this study. I wonder if the brachiopod specific clumped isotope equation proposed by the authors is significantly different from Meinicke et al. if projected on the I-CDES scale. If so, it might not be warranted to propose a new clumped isotope calibration as brachiopods might be calibrated with general calcite calibrations.

Meinicke, N., Ho, S. L., Hannisdal, B., Nürnberg, D., Tripati, A., Schiebel, R., and Meckler, A. N.: A robust calibration of the clumped isotopes to temperature relationship for foraminifers, Geochimica et Cosmochimica Acta, 270, 160–183, https://doi.org/10.1016/j.gca.2019.11.022, 2020.

Meinicke, N., Reimi, M. A., Ravelo, A. C., and Meckler, A. N.: Coupled Mg/Ca and Clumped Isotope Measurements Indicate Lack of Substantial Mixed Layer Cooling in the Western Pacific Warm Pool During the Last ~5 Million Years, Paleoceanography and Paleoclimatology, 36, e2020PA004115, https://doi.org/10.1029/2020PA004115, 2021.

de Winter, N. J., Witbaard, R., Kocken, I. J., Müller, I. A., Guo, J., Goudsmit, B., and Ziegler, M.: Temperature Dependence of Clumped Isotopes (âˆ†47) in Aragonite, Geophysical Research Letters, 49, e2022GL099479, https://doi.org/10.1029/2022GL099479, 2022.

For a detailed response to this comment we refer the reader to "Reply on RC1" of the interactive discussion. To make things short, there is a scientific disagreement between the referee and us (the authors). For reasons detailed in "Reply on RC1", we do not consider that the Anderson et al. (2021) equation induces a cold bias. Rather we consider that significant deviation of $\Delta_{47}$ values from the Anderson et al. (2021) equation reflect other processes than temperature that controls $\Delta_{47}$ values in brachiopod shell calcite (Bajnai et al., 2018) and other carbonates (Bajnai et al., 2020). However, we agree that to propose a brachiopod specific equation is not a good way to circumvent the observed discrepancies especially as the deviations from the Anderson et al. (2021) equation do not concern all of the specimens studied here.

Lines 435-464: The discussion over the clumped isotope results was significantly reworked. Especially, we highlight that the deviation from the Anderson et al. (2021) equation are not of the same magnitude in different brachiopod groups, which likely relate to biologic processes yet to confidently identified.

**Minor comments**

Line 31: "but is significantly offset" rephrase to "are significantly offset" ("D47 values" is plural)

Line 31: rephrased as "(our data) confirm significant offsets from …"

Line 102-104: Perhaps the authors could add here that current empirical clumped isotope temperature calibrations also show good agreement with *ab initio* models of the carbonate isotope system, as was demonstrated in Jautzy et

al. (2020). This is another argument in favor of the use of common clumped isotope thermometers in a variety of carbonate materials.

Jautzy, J. J., Savard, M. M., Dhillon, R. S., Bernasconi, S. M., and Smirnoff, A.: Clumped isotope temperature calibration for calcite: Bridging theory and experimentation, Geochemical Perspectives Letters, 14, 36–41, 2020.

This precision and reference was added lines 104-105

Line 167: "graving bit" should this read "engraving bit"?

Corrected accordingly

Line 302-305: "At temperate and polar temperatures (20 to 0°C) our equation has a steeper slope than that of Brand et al. (2019)" The authors need to explain this in more detail and/or refer to a figure where the reader can spot this effect. The authors refer to figure 2A later, but it is not clear which line in this figure represents the equation by Brand. I do not understand how the slope can be different with temperature if a linear equation (with a constant slope) is compared, but I might misunderstand what the authors are trying to say. Later on, the authors mention that the Brand et al. equation is non-linear, but I still have trouble following the description of the comparison in this section.

Another referee also stated that this part was unclear. We will address this issue in the revised manuscript and provide further and clearer discussion regarding the difference in "slope" of different equations.

Lines 302-303: This part was rephrased.
This aspect is further illustrated by the comparison of the fractionation equations added in the revised version (lines 378-386; new Figure 4)

Line 408: "In consequence" rephrase to "As a consequence"

Corrected

Line 514: "constrain" should read "constraint"

Corrected

Line 515: "should be privileged for trace-element-based paleotemperatures reconstructions" consider rephrasing to "should be selected/prioritized for trace element based paleotemperature reconstructions"

Changed to "preferred"

Line 550-567: This clear grouping based on trace element content is an interesting observation. I wonder if the authors considered whether there might be a relationship with growth rate. Does the "high" or the "low" group show significantly faster growth than the other? If so, this could be an explanation for the difference in shell composition, as trace element concentrations in calcifiers often show a correlation with growth rate. In addition, in this section about grouping of specimens based on trace element content, adding a figure showing the differences in concentration would be helpful.

This trend may be related to growth rates. Unfortunately, our new dataset comprises only 5 species for which we have constraints on their growth rate, limiting quantitative comparisons. This hypothesis could be mentioned into the discussion. A figure will be added to illustrate this dichotomy. We must also note that this difference is confined to

the inner shell layers. The trends described here between Terebratellidina and Terebratulidina disappear while looking at the outer shell layers.

The discussion was reworked and the grouping is highlighted in the new figure 6.
Because we have growth rate constraints for only 5 species we did not relate this directly to growth rate, but we discuss this grouping in section 4.2 in relation to the kinetic effects highlighted in the oxygen isotopes and clumped isotopes (lines 657-695).

Line 606: The authors might consider rephrasing the title of this section to: "Precipitation of brachiopod shell calcite out of equilibrium with seawater"

Title rephrased accordingly.

Line 626-630: The authors might consider citing the recent study by Garbelli et al. (2022) here who also interpret changes in isotopic composition of (fossil) brachiopod shells as seasonal variability.

Garbelli, C., Angiolini, L., Posenato, R., Harper, E. M., Lamare, M. D., Shi, G. R., and Shen, S.: Isotopic time-series ($\delta13C$ and $\delta18O$) obtained from the columnar layer of Permian brachiopod shells are a reliable archive of seasonal variations, Palaeogeography, Palaeoclimatology, Palaeoecology, 607, 111264, https://doi.org/10.1016/j.palaeo.2022.111264, 2022.

We agree that the addition of literature regarding seasonal variability registered in brachiopod shells would be pertinent here. Although very interesting and convincing, the work of Garbelli et al. (2022) may not be the most pertinent reference here, when other papers studying modern brachiopods in monitored environment unambiguously highlight seasonal variability (Yamamoto et al., 2011; Takayanagi et al., 2015).

Added a call to the references suggested in response.

Lines 653-673: It seems that some of the discussion of kinetic (growth rate-related) effects in trace element composition could be a useful addition to the section above where the observation of differences in trace element composition between brachiopod groups is discussed (see my comment on lines 443-466).

This trend may be related to growth rates. Unfortunately, our new dataset comprises only 5 species for which we have constraints on their growth rate, limiting quantitative comparisons. This hypothesis could be mentioned into the discussion. A figure will be added to illustrate this dichotomy. We must also note that this difference is confined to the inner shell layers. The trends described here between Terebratellidina and Terebratulidina disappear while looking at the outer shell layers.

The discussion over kinetic effects and their likely impact on trace element content was reworked. The hypothesis of a kinetic control on trace element incorporation is more clearly stated and element/Ca ratios are further compared with isotopic deviation from equilibrium which we associated at first order to kinetic effects.

Line 681: "isotopic" should read "isotopic"

Section rewritten

Line 688-689: "Given the highly…may be coincidental." I think the authors should explain this line of reasoning in a bit more detail.

Here we intend to provide a critical look at our dataset as the differences in both trace elements and clumped isotopes may be explained by two kinds of groupings in our dataset: 1)High latitudes vs low latitudes with difference in seasonality of the ecosystem that may induce different growth rate dynamics and 2) taxonomic grouping which may involve differences in shell formation processes including growth rate. However, as the Terebratulidina are mostly associated with low latitudes and Terebratellidina with mid-high latitudes in the dataset, we lack strong arguments to prefer a hypothesis over the other.
We will rephrase this part to make it easier to understand.

This was rephrased (lines 679-680).
The issue of a latitudinal among the different Terebratulida suborders is clearly highlighted in the new Figure 6 C and D.

Line 736: Please add the missing "delta" before "$_{18}O_{sw}$".

Section removed

Line 740: "central values" is a bit of a cryptic term, perhaps the authors mean median or mean/average values?

Section removed

Lines 710-746: I think the addition of a fossil case study is nice, but it is not essential for the study. If the authors would like to keep their manuscript more concise, this is a section that could be significantly shortened or removed in my opinion.

Another referee has done a similar comment. This section will be removed from the revised version.

Section removed

Table 4: As mentioned in one of my previous comments, it might be worthwhile to add the clumped isotope-based temperature reconstructions based on the Meinicke et al. calibration, since these are likely more accurate and more closely in line with the modern brachiopod data in this study as well.

Section removed

Line 796-797: The temperature underestimation by ~3 degrees is very similar to the offset found in de Winter et al. (2022; see comment above) and this nice corroboration between multiple datasets is worth mentioning.

Line 797-798: As mentioned above, I tend towards disagreeing with this call for a brachiopod-specific clumped isotope calibration, since most of the offset in clumped isotope values may be explained by the Anderson et al. equation underestimating temperatures in general (not just for brachiopods). The authors should consider this explanation before suggesting a taxon-specific calibration is needed.

Response to both comments: We retracted from the brachiopod specific equation and rather highlight that brachiopod shells $\Delta_{47}$ values can significantly depart from isotopic equilibrium and propose likely explanations.

**Comments from: Adrian Immenhauser (Referee)**

Abstract:

Poorly written. Many deficits in the logic and precision of the language. I list several points that caught my attention; there are others.

Ln 13. Please use `seawater´ rather than `ocean'. Delta18O measurements. That is jargon. A `measurement' is the analytical step(s) we perform to generate the data, the analysis so to speak. The reconstruction of past seawater properties is based on oxygen isotope data (not the measurement thereof).

Corrected

Ln. 18. Missing word: commonly used `archives' in studies….

Corrected

Ln. 19. Unclear wording: …resistant to diagenetic alteration for decades. Do you mean brachiopod shells do not alter over the time span of several decades? Or do you mean that over the past decades, scientists have considered brachiopod shells to be resistant to diagenetic alteration?

Rephrased

Ln. 21. I am not sure what a `growing temperature' is (also referred to as `living temperature' elsewhere)? I did google the term to make sure I did not miss something. The only paper that matched is yours (this discussion version) on the EGU sphere webpage. Do you mean the ambient seawater's temperature during the brachiopod's lifetime? Use `ambient seawater temperature', I suggest. Other than that, please use proper terminology: seawater $d_{18}O$ `values' or similar.

'Growing temperature' which may be corrected as 'growth temperature' or 'shell growth temperature', refers to the ambient temperature when the animal grows its shell. This formulation aims to point that the temperature that may be registered in the carbonate archive, will correspond to the temperature of the environment when the carbonate formed. This exclude the periods during which the animal live but does not grow its shell. We acknowledge that in this specific occurrence the use of 'seawater temperature' is better suited.

Replaced by "seawater temperature". Correct mentions to "$\delta^{18}O$ values" has been checked throughout the text.

Ln. 23. Again, I can only guess what a `supposed´ carbonate-based palaeothermometre is. Do you mean `novel´ or `less well established´? Moreover, the palaeothermometre is NOT based on carbonate but uses the archive data (geochemical properties) recorded in carbonate (note the difference between the terms archive and proxy).

Rephrased

Ln. 31. Missing word: …with `seawater´ temperatures…

Rephrased

Ln. 37. What do you mean by `relatively good´. In agreement with the measured temperatures within xy degree Celsius?

'good' should be here replaced by 'strong' relative to the strength of the correlation between the geochemical parameters and the temperature, which is illustrated by the regression coefficient.

Rephrased

Introduction:
Much better written, but still some language deficits similar to those listed for the case of the abstract. Please consider.

Ln. 48. What are past seas, and what is the difference to oceans? Do you mean epeiric seas as opposed to genuine oceanic bluewater?

That is what we mean. This difference is here to imply that the $\delta^{18}O$ values of the seas that may be more or less restricted, can deviate substantially from that of the open ocean. Especially as while $\delta^{18}O$ values of the open ocean mostly reflect global processes (Amount of continental ice, oceanic circulation, global climate), the $\delta^{18}O$ values of the seas are also influenced by more local processes (runoff, evaporation).

This was kept in the corrected version.

Ln. 62. Many inconsistencies concerning technicalities of cited references. See, for example, ln. 52. Brand et al., (2013) should read Brand et al. (2013).

Corrected. Similar mistakes have been checked throughout the manuscript.

Ln. 121. That is a scientific criticism. The authors argue about the question of whether shell carbon (DIC) and oxygen isotope values are in equilibrium with the seawater from which the shell carbonate precipitated or not. Please allow me to clarify that brachiopod biominerals are secreted from bodily fluids, NOT seawater. The problem is threefold: (i) What is the isotopic value of the bodily fluid relative to that of the ambient seawater? (ii) Does the isotopic value of the bodily fluid change during active versus passive cycles in the brachiopod metabolism cycle and during the brachiopods life span? Juvenile brachiopods grow rapidly, mature slow down. (iii) What is the fractionation factor between bodily fluid and brachiopod biomineral, and is it constant during the lifetime of a brachiopod? In some cases, brachiopod bodily fluids are isotopically close to the ambient seawater; in others not. In short, it is complicated. The authors provide text about thermodynamics and kinetics but less so about these metabolic effects and biomineralization pathways. In my oppinion, that is a weakness of the paper. Please see the discussion and references cited in:

Immenhauser, A., Schöne, B., Hoffmann, R. and Niedermayr, A. (2016) Mollusc and brachiopod skeletal hard parts: Intricate archives of their marine environment. Sedimentology 63, 1-59.

I emphasise that you do not need to cite my paper! That is entirely up to you. It simply saves the reviewer time when being able to refer to the text and the cited references in a published paper. Please consider.

We fully agree that rynchonelliform brachiopods shells are not the result of an inorganic precipitation experiments from the seawater, but that they result from biological processes promoting carbonate precipitation from a biologically controlled fluid, forming a structured carbonate shells where calcite crystals are embedded in an organic matrix (Williams, 1968; Curry et al., 1991; Gaspard et al., 2008; Simonet Roda et al., 2019, 2022). We acknowledge that these aspects are not very present in the paper, mostly as the scope of this paper is the use of brachiopod shell geochemistry for paleo-environment reconstruction. But we are fully aware of the differences between biogenic and inorganic carbonate precipitation and will put more emphasis on these aspects in the revised version. From a more methodological point of view, the comparison between the brachiopod shell mineralisation and inorganic calcite precipitation is an approach to highlight the effect of biological processes on shell geochemistry, especially when the

chemistry of the mineralizing fluids for brachiopods, remains for now largely unknown (We do not know of any study reporting isotopic values of ionic concentration from the brachiopod body fluids).

We added to the introduction several aspects on brachiopod shell formation to highlight what biological and environmental processes may affect brachiopod calcite precipitation and its chemistry. Line (108-120)

Material and Methods:

No major comments, looks o.k. One exception, please avoid acronyms in titles (2.3) and
please refer to `values'. $d_{13}C$ of modern brachiopods is jargon. Please use carbon isotope values of modern….

Title rephrased (line 191)

Results:

Header chapter 3.1. Please do not use `stable' as a synonym for carbon and oxygen
isotope values; science knows about 120 stable isotopes.

Line (244): Header rephrased

Ln. 245. I always wonder, what is the meaning of the second decimal in a range of isotope values resulting from bulk samples? What is the meaning of -2.24 permil in this context? My opinion, the second decimal is meaningless. You analyse a bulk sample from a brachiopod shell, and I would refer to that as pseudo-precision. I suggest providing one decimal values. Bulk samples and second decimals do not match. Particularly as you mix bulk sample data and data from the inner and outer shells (see Table 2).

$\delta^{13}C$ and $\delta^{18}O$ values reported with only one decimal in text and tables.

Discussion:

General comment: This chapter is longwinded and, in part, difficult to follow. I advise streamlining the text and shortening it by at least 20%. I wonder if the `holy trilogy´ of scientific writing consisting of Data Presentation, Data Interpretation and Discussion is applied here? If so, where is the discussion? Consider rephrasing the header as `Interpretation and Discussion'.

Section 4.3. was removed. Some parts were rewritten to clarify the discussion. However, there is some added discussion in order to answer the comments of other referees. Header was rephrased.

Ln. 333. What are `independent´ brachiopods? Please explain.

Obviously here we do not refer to brachiopods that are independent from anyone or anything. 'Independent' here qualifies the dataset chose to test the fractionation equations. The adjectives associated with dataset here are numerous so we will rephrase it to make it clearer.

Rephrased

Chapter 4.1.2 is poorly written. Quite some problems regarding grammar and formalities
(citations etc.). Please clean up.

This section has been significantly reworked (lines 414-464)

Chapter 4.1.4 All good science but very longwinded. Could you streamline that? This is not easy to follow and this is not something you want to hear from the readers.

We will rework the construction of this section to make it easier to follow and more concise.

The last paragraph of this section was reworked in depth (lines 539-589) with more call to the data, and an added figure (Figure 6) as asked by anonymous referee 1, which should help follow this discussion.

Chapter 4.2 Here, we need much more emphasis on metabolism and biomineral secretion from bodily fluids. The authors deal with the topic as if brachiopod biomineralization pathways were an inorganic precipitation experiment. These are super complicated little bio-machines', and they are fascinating since each individual is a case on its own. Please see papers from the marine biology community (mainly aquaria monitoring experiments but also field observations).

Indeed, if the data can be explained by what is known of inorganic precipitation, then we do not need to invoke any biological processes. On the contrary if the data diverge from what we expect from inorganic precipitation, then biological processes may explain these differences.
Regarding the discussion around kinetic effects, we clearly state (ln 689-709) that we expect the extent of the kinetic effects to be directly linked to shell growth rates, which is biologically controlled.

More emphasis was put on biological processes in this sections and others. There is a focus on kinetic effects which can be related to shell growth rates and which we identify here. We invoke other possible sources of biologic or environmental, but that our dataset is unable to resolve.

Conclusion(s):

Please use the plural, I suggest that you list more than one conclusion here.
This chapter is very much written in a discussion style. Please consider coming up with genuine conclusions style text rather than a short (renewed) discussion. The last statement is an anti-climax. First, you present all of these data and text. Then you tell the reader that you advise considering the variability in brachiopod live habitats, environmental conditions, metabolic effects, seasonal effects etc.? I must admit, have read very similar concluding statement in many papers published a decade or more years ago. Please consider.

Header corrected. The conclusions were almost completely rewritten.

**Comments from Anonymous Referee #3**

Comparison of the new equation of the oxygen isotope fractionation with those of Brand et al. (2019). Both equations are similar in the temperature range between 10-25°C, but differ in the low-temperature field (<10°C). Since the Brand et al brachiopod-data set is by far much better constrained by data points, the authors should provide a more in-depth discussion about causes of the offset. Further, they should strengthen their arguments why it is necessary to introduce the new equation, and why it shall be an improvement.

We will clarify this part and add elements of comparison between the different fractionation equations.

We discuss an example of the possible source of discrepancies between the datasets, and highlight differences in the environmental parameters (Temperature, $\delta^{18}O_{sw}$ values) used to determine the equation (lines 313-322).
The comparison of the different equations by applying them to the dataset of Bajnai et al. (2018) was pushed further by comparing the distribution of the temperature offsets with growing temperature (lines 378-386; new figure 4)

In this context, please also indicate the MAT range covered by the Bajnaj et al. 2018 brachiopod data set (line 335).

MAT ranged added

The supposed sampling procedure avoids specialized parts of the shell as umbo, edges, muscle scars, primary layers. However, Ullmann et al. (2017) observed additional significant taxon-specific ranges in their intra-specific high-resolution oxygen isotope data. How has this observation an effect for the results of this study? Is intra-specific variability smoothed by the sample size? Which degree of uncertainty introduce specimens of the suborder Terebratellidina to the fractionation equations? Please comment on this.

We assume that the sampling area, which spans at least two major growth lines, allows a smoothing of the intra-specimen variability within the collected homogenized powder. While major growth lines are commonly interpreted as annual, that is not necessarily the case, which explains the uncertainty on the time represented by the sampling area. This sampling creates mass-averaged geochemical compositions, which is mathematically different from time-averages which we use for environmental conditions, because of varying shell growth rates through time. The more variable the growth rate, the more biased this sampling will be towards periods of high growth rates (Schöne, 2008; Yamamoto et al., 2011; Takayanagi et al., 2015).
Specimens of the suborder Terebratellidina highlight clumped isotope deviations from equilibrium and trace elements compositions of their inner layer significantly different than specimens form the suborder Terebratulidina. However, deviation from thermodynamic equilibrium in oxygen isotope composition are not significantly different between the two groups, within the dataset tested. So we do not expect the fractionation equation to be biased by any of the groups.

The Jurassic example is not well executed and not the scope of this study. New and very few data points are introduced first time in the discussion. I recommend the removal of this part of the manuscript, since its focus is on modern brachiopod taxa.

As also suggested by another referee, this part will be removed from the revised version of the manuscript

Part removed from the revised version.

Please, explain all parameters and abbreviations in Supplementary Table S1.

All columns are described in more details

Please check the manuscript for spelling errors, here are some I spotted:

"Rhynchonellida" in Fig. 4 – revise spelling two "l"

Figure corrected

Line 103: Delete "previous"

Deleted

Line 129-131: incomplete sentence

The sentence appears complete.

Line 375: Enter a space between "regression derived"

corrected

Line 532: spelling of "isotopic fractionation"

Corrected

Typos in supplementary File S3 (Sheet Description)

Corrected

**References:**

Bajnai, D., Fiebig, J., Tomašových, A., Milner Garcia, S., Rollion-Bard, C., Raddatz, J., Löffler, N., Primo-Ramos, C., and Brand, U.: Assessing kinetic fractionation in brachiopod calcite using clumped isotopes, Sci. Rep., 8, 533, https://doi.org/10.1038/s41598-017-17353-7, 2018.

Bajnai, D., Guo, W., Spötl, C., Coplen, T. B., Methner, K., Löffler, N., Krsnik, E., Gischler, E., Hansen, M., Henkel, D., Price, G. D., Raddatz, J., Scholz, D., and Fiebig, J.: Dual clumped isotope thermometry resolves kinetic biases in carbonate formation temperatures, Nat. Commun., 11, 4005, https://doi.org/10.1038/s41467-020-17501-0, 2020.

Curry, G. B., Cusack, M., Walton, D., Endo, K., Clegg, H., Abbot, G., and Armstrong, H.: Biogeochemistry of brachiopod intracrystalline molecules, Philos. Trans. R. Soc. Lond. B. Biol. Sci., 333, 359–366, https://doi.org/10.1098/rstb.1991.0085, 1991.

Gaffey, S. J. and Bronnimann, C. E.: Effects of bleaching on organic and mineral phases in biogenic carbonates, J. Sediment. Res., 63, 752–754, https://doi.org/10.1306/D4267BE0-2B26-11D7-8648000102C1865D, 1993.

Gaspard, D., Marie, B., Luquet, G., and Marin, F.: Biochemical BlackwellPublishingLtd characteristics of the soluble organic matrix from the shell of three Recent terebratulid brachiopod species, 2008.

Key, M. M., Smith, A. M., Phillips, N. J., and Forrester, J. S.: Effect of removal of organic material on stable isotope ratios in skeletal carbonate from taxonomic groups with complex mineralogies, Rapid Commun. Mass Spectrom., 34, https://doi.org/10.1002/rcm.8901, 2020.

LeGrande, A. N. and Schmidt, G. A.: Global gridded data set of the oxygen isotopic composition in seawater, Geophys. Res. Lett., 33, L12604, https://doi.org/10.1029/2006GL026011, 2006.

Pingitore, N. E., Borrego, P. M., and Crawford', G. M.: Dissolution kinetics of CaCO3 in common laboratory solvents, J. Sediment. Petrol., 63, 641–645, 1993.

Schöne, B. R.: The curse of physiology—challenges and opportunities in the interpretation of geochemical data from mollusk shells, Geo-Mar. Lett., 28, 269–285, https://doi.org/10.1007/s00367-008-0114-6, 2008.

Schöne, B. R., Schmitt, K., and Maus, M.: Effects of sample pretreatment and external contamination on bivalve shell and Carrara marble δ18O and δ13C signatures, Palaeogeogr. Palaeoclimatol. Palaeoecol., 484, 22–32, https://doi.org/10.1016/j.palaeo.2016.10.026, 2017.

Simonet Roda, M., Ziegler, A., Griesshaber, E., Yin, X., Rupp, U., Greiner, M., Henkel, D., Häussermann, V., Eisenhauer, A., Laudien, J., and Schmahl, W. W.: Terebratulide brachiopod shell biomineralization by mantle epithelial cells, J. Struct. Biol., 207, 136–157, https://doi.org/10.1016/j.jsb.2019.05.002, 2019.

Simonet Roda, M., Griesshaber, E., Angiolini, L., Rollion-Bard, C., Harper, E. M., Bitner, M. A., Milner Garcia, S., Ye, F., Henkel, D., Häussermann, V., Eisenhauer, A., Gnägi, H., Brand, U., Logan, A., and Schmahl, W. W.: The architecture of Recent brachiopod shells: diversity of biocrystal and biopolymer assemblages in rhynchonellide, terebratulide, thecideide and craniide shells, Mar. Biol., 169, 4, https://doi.org/10.1007/s00227-021-03962-4, 2022.

Takayanagi, H., Asami, R., Otake, T., Abe, O., Miyajima, T., Kitagawa, H., and Iryu, Y.: Quantitative analysis of intraspecific variations in the carbon and oxygen isotope compositions of the modern cool-temperate brachiopod Terebratulina crossei, Geochim. Cosmochim. Acta, 170, 301–320, https://doi.org/10.1016/j.gca.2015.08.006, 2015.

Williams, A.: A history of skeletal secretion among articulate brachiopods, Lethaia, 1, 268–287, https://doi.org/10.1111/j.1502-3931.1968.tb01741.x, 1968.

Yamamoto, K., Asami, R., and Iryu, Y.: Brachiopod taxa and shell portions reliably recording past ocean environments: Toward establishing a robust paleoceanographic proxy: BRACHIOPOD OXYGEN ISOTOPE RECORDS, Geophys. Res. Lett., 38, L13601, https://doi.org/10.1029/2011GL047134, 2011.

York, D., Evensen, N. M., Martínez, M. L., and De Basabe Delgado, J.: Unified equations for the slope, intercept, and standard errors of the best straight line, Am. J. Phys., 72, 367–375, https://doi.org/10.1119/1.1632486, 2004.

---

## Author Response (AR2)

**Authors response to Anonymous Referee #1 report #1**

Dear Prof. Tina Treude, dear referee, dear EGU sphere,

Here is our response to the comments of Anonymous Referee #1 in the last report and to the Associate Editor decision.

First, we answered the Referee suggestion and Associate Editor decision of adding a comparison of our data with the equation of Meinicke et al., (2021). However, for reasons stated in detail bellow, we extended this comparison also to the equations of Peral et al. (2022) and Huyghe et al. (2022) which also use low temperature (<30°C) datasets anchored to the I-CDES reference frame. This comparison highlight that a number of modern brachiopod calcite $\Delta_{47}$ values not only deviate from the equation of Anderson et al. (2021) but also form published equations based only on biogenic marine carbonates (foraminifera, bivalves). The detailed results of this comparison are made available in an updated version of Supplement S3.

We additionally corrected the values of $\Delta_{47}$ temperature offsets for Terebratellidina and Terebratulidina (Section 4.1.2). Values in previous versions of the manuscript resulted from a calculation with one sample attributed to the incorrect taxonomic group. This change has no impact on our conclusions. All these changes relate to Section 4.1.2, pages 18-19 in the revised manuscript.

We also take the opportunity that this document will be public to further answer in details the arguments advanced by the referee in the last report, regarding the potential cold bias of the Anderson et al. (2021) clumped isotope temperature equation.

Referee coment

Author's response

That said, I believe the scientific disagreement between the authors and myself regarding a potential cold bias of the Anderson D47 equation in the lower temperature range has not been fully resolved. I appreciate the detailed reply to this issue the authors gave in the online discussion. I also fully agree that the way "true" temperatures are assigned to the foraminifera datapoints in the Meinicke, Piasecki and Peral datasets is not ideal. This is, in my opinion, not a fault of the authors, but a general issue with calibration datasets based on microfossils (unless grown in lab cultures).

We fully agree on that last point.

However, I remain of the opinion that the fact that data from lab-grown bivalve shells (with highly precise known temperatures) does align significantly better with the Meinicke dataset than the Anderson dataset should be a good reason for the authors to at least consider the reprocessed Meinicke equation in their analysis. The fact that data from mollusk shells, which are known to be precipitated close to equilibrium (as acknowledged by the author and

demonstrated in Huyghe et al. 2022) is significantly offset from the Anderson line in the same direction as the brachiopod data in this study should also be mentioned in my view.

We acknowledge that some of the clumped isotope data from de Winter et al. (2022) are offset from the Anderson et al. (2021) equation in the same direction as our brachiopod data. The fact that these results align better with the equation of Meinicke et al. (2021) is however, only relevant to the *Arctica islandica* samples grown at 15 and 18°C, according to table 1 in de Winter et al. (2022). The better agreement with the Meinicke et al. (2021) equation rather than the Anderson et al. (2021) equation is not significantly relevant to the rest of the low temperature aragonite dataset according to de Winter et al. (2022) themselves: "*When including clumped isotope values of other low-temperature (<30°C) aragonites in the compilation, the regression remains indistinguishable from the calibration of Anderson et al. (2021) and similar to the foraminifera-based calibration by Peral et al. (2018) and Meinicke et al. (2020) combined with reference to I-CDES in Meinicke et al. (2021) and the Guo et al. (2009) theoretical temperature relationships (Figure 2b)*" (Section 3.2 in de Winter et al., 2022). This statement is supported by their supplementary data S8, where low temperature (<30°C) aragonite $\Delta_{47}$ values from their compilation show no statistically significant offset with the equation of Meinicke et al. (2020), neither with that of Anderson et al. (2021). In consequence there is no strong argument to support that either of these equations is significantly different or better than the other to characterize the low temperature aragonite dataset.

While we acknowledge the precise control over temperature in lab grown bivalves, this does not rule out any other control over shell geochemistry, especially biological controls that may create deviation from equilibrium. Indeed, while *Arctica islandica* precipitate its shell close to oxygen isotope equilibrium with seawater, it is not the case for carbon isotopes or trace elements (see the review of Schöne, 2013 on *Arctica islandica* and ref therein). In addition, while the dataset of Huyghe et al. (2022) highlight $\Delta_{47}$ values from bivalve calcite close to the expected equilibrium, to generalize this observation to all mollusc shells could be faulty, especially at light of the disequilibrium reported in juvenile oysters also by Huyghe et al. (2022).

With these considerations, the offset observed in *Arctica islandica* $\Delta_{47}$ values relative to the Anderson et al. (2021) equation could well be explained by other processes including biologic processes, rather than a bias in the equation. Indeed, Figure 1 and Table 1 of de Winter et al. (2022) shows that *Arctica islandica* $\Delta_{47}$ values of specimens grown at temperatures of 3.2 and 1.1°C are in agreement with the equation of Anderson et al., (2021) while specimens grown at temperatures of 15 and 18°C show significant deviations from this equation. Again from the review of Schöne (2013) several aspects of *Arctica islandica* physiology may explain such deviation.

_ "*This species tolerates temperature and salinity ranges of 1° to 16 °C (Golikov and Scarlato, 1973; Mann, 1989; Witbaard et al., 1997a; tolerance under experimental conditions for limited amounts of time up to 20 °C: Winter, 1969) and 22 to 35 PSU (Winter, 1969; Oeschger and Storey, 1993), respectively*" (Schöne, 2013)

_ "*Shell growth occurs at temperatures as low as 1 °C, increases strongly between 1° and 6 °C and shows a tenfold increase between 1° and 12 °C (Witbaard et al., 1997b).*" (Schöne, 2013)

We argue that a significant increase in shell growth rate between the specimens grown at 1 and 3°C and those grown at 15 and 18°C may result in kinetic effects similar to the ones suggested for brachiopod shells (Bajnai et al., 2018; this study). This hypothesis could be easily tested with constrains on shell growth which was monitored during the experiment according to Supplementary S1 from de Winter et al. (2022), but we were not able to find that information.

Finally, though admittedly slightly off-topic, I am not convinced that the author's argument (made in reply to my comment in the online discussion) that several D47 datasets agree within uncertainty with the Anderson dataset is very strong in this context, because the potential cold bias is only argued for by de Winter and colleagues in the cold end of the temperature range, and many of these datasets (e.g. Anderson and Jautzy) contain many hot datapoints that can strongly influence the slope of a linear regression (especially if the actual temperature relationship may not be linear).

The Anderson et al. (2021) equation is very well constrained by data points in the low temperature range stated by de Winter et al. (2022) (<30°C) by the reprocessed foraminifera datasets of Breitenbach et al. (2018), Peral et al. (2018) and Meinicke et al. (2020), by the very slow growing mammillary calcite of Laghetto Basso (8°C) (Anderson et al., 2021; Fiebig et al., 2021), by calcite from perennial ice-covered lakes (Anderson et al. 2021). Plus, the bivalve calcite dataset of Huyghe et al. (2022) further support the Anderson et al. (2021) equation in this low temperature range. That is what we illustrated in the figure in our previous response and is illustrated by Figure 3B in Anderson et al., (2021). Thus multiple datasets, some of them with very strong temperature control, argue against a cold bias of Anderson et al. (2021) in the low temperature range (<30°C). The bivalve aragonite data reported by de Winter et al. (2022) and some of brachiopod calcite data of Bajnai et al. (2018) and our study are for now only a few deviations from Anderson et al. (2021) equation relative the quantity of data that support that equation in the low temperature range.

Alternatively, we also raise the following question. In the hypothesis that the Anderson et al. (2021) has a cold bias related to the use of high temperature points in the regression, is there any reason to prefer the Meinicke et al. (2021) equation over other equations derived in marine temperature range and set in the I-CDES reference frame such as the equation of Peral et al. (2022) or that of Huyghe et al. (2022) ?

This is why in the revised version we compare our results not only to the equation of Meinicke et al. (2021) but also to that of Peral et al. (2022) and that of Huyghe et al. (2022) which do not include high temperature data points. The application of those two last equations to the brachiopod dataset result in similar cold bias as using the equation on Anderson et al. (2021), while applying the equation of Meinicke et al. (2021) only slightly reduces the cold bias in our dataset. Not only does this comparison discard the argument that the observed cold bias result from the regression of Anderson et al. (2021) equation with high temperature datapoints, it highlights that none of the published equation can fully resolve the cold bias observed in brachiopod shells.

In addition, the Anderson dataset includes common data points with most of the other datasets mentioned in the author's reply.

Indeed, as the Anderson et al. (2021) equation is a composite equation based on multiple previously published datasets. It also includes the Meinicke et al. (2020) foraminifera dataset albeit with different calcification temperature assignment as already discussed.

To sum up over the case of the Meinicke et al. (2021) vs Anderson et al. (2021) equations:

1) There is in our opinion no significant reason in the study of de Winter et al. (2022) to prefer either of the Meinicke et al. (2021) or Anderson et al. (2021) equation to describe aragonite $\Delta_{47}$ values.

2) There is no influence of the non-linearity of the T-D47 that result in a cold bias of the Anderson et al. (2021) equation in the low temperature range (<30°C). Indeed, the Anderson et al. (2021) composite equation is very well constrained by data points with temperature <30°C, plus, similar cold biases on the brachiopod dataset are observed applying equations constrained by marine samples (Peral et al., 2022; Huyghe et al. 2022).

3) We propose a hypothesis to explain the deviation of *Arctica islandica* $\Delta_{47}$ values from the Anderson et al. (2021) equation reported by de Winter et al. (2022), and call for a test of this hypothesis if possible.

Finally, with all these elements, as the Anderson et al. (2021) equation is a composite of datasets from various labs and various carbonate origins, it appears, to date, to be the best constrained equation to describe $\Delta_{47}$-Temperature relationship in calcium carbonate materials (calcite and aragonite). Thus, it should be preferred to other equations with less data constrains such as those of Meinicke et al. (2021), Huyghe et al. (2022) and Peral et al. (2022). Nevertheless, this conclusion may be revised in a near future with increasing number of data reporting significant deviations from this equation (Bajnai et al., 2020; Fiebig et al., 2021; de Winter et al., 2022; this study)

**References**

Anderson, N. T., Kelson, J. R., Kele, S., Daëron, M., Bonifacie, M., Horita, J., Mackey, T. J., John, C. M., Kluge, T., Petschnig, P., Jost, A. B., Huntington, K. W., Bernasconi, S. M., and Bergmann, K. D.: A Unified Clumped Isotope Thermometer Calibration (0.5–1,100°C) Using Carbonate-Based Standardization, Geophys. Res. Lett., 48, https://doi.org/10.1029/2020GL092069, 2021.

Bajnai, D., Fiebig, J., Tomašových, A., Milner Garcia, S., Rollion-Bard, C., Raddatz, J., Löffler, N., Primo-Ramos, C., and Brand, U.: Assessing kinetic fractionation in brachiopod calcite using clumped isotopes, Sci. Rep., 8, 533, https://doi.org/10.1038/s41598-017-17353-7, 2018.

Bajnai, D., Guo, W., Spötl, C., Coplen, T. B., Methner, K., Löffler, N., Krsnik, E., Gischler, E., Hansen, M., Henkel, D., Price, G. D., Raddatz, J., Scholz, D., and Fiebig, J.: Dual clumped isotope thermometry resolves kinetic biases in carbonate formation temperatures, Nat. Commun., 11, 4005, https://doi.org/10.1038/s41467-020-17501-0, 2020.

Breitenbach, S. F. M., Mleneck-Vautravers, M. J., Grauel, A.-L., Lo, L., Bernasconi, S. M., Müller, I. A., Rolfe, J., Gázquez, F., Greaves, M., and Hodell, D. A.: Coupled Mg/Ca and clumped isotope analyses of foraminifera provide consistent water temperatures, Geochim. Cosmochim. Acta, 14, 2018.

Fiebig, J., Daëron, M., Bernecker, M., Guo, W., Schneider, G., Boch, R., Bernasconi, S. M., Jautzy, J., and Dietzel, M.: Calibration of the dual clumped isotope thermometer for carbonates, Geochim. Cosmochim. Acta, 312, 235–256, https://doi.org/10.1016/j.gca.2021.07.012, 2021.

Huyghe, D., Daëron, M., de Rafelis, M., Blamart, D., Sébilo, M., Paulet, Y.-M., and Lartaud, F.: Clumped isotopes in modern marine bivalves, Geochim. Cosmochim. Acta, 316, 41–58, https://doi.org/10.1016/j.gca.2021.09.019, 2022.

Meinicke, N., Ho, S. L., Hannisdal, B., Nürnberg, D., Tripati, A., Schiebel, R., and Meckler, A. N.: A robust calibration of the clumped isotopes to temperature relationship for foraminifers, Geochim. Cosmochim. Acta, 270, 160–183, https://doi.org/10.1016/j.gca.2019.11.022, 2020.

Meinicke, N., Reimi, M. A., Ravelo, A. C., and Meckler, A. N.: Coupled Mg/Ca and Clumped Isotope Measurements Indicate Lack of Substantial Mixed Layer Cooling in the Western Pacific Warm Pool During the Last ~5 Million Years, Paleoceanogr. Paleoclimatology, 36, https://doi.org/10.1029/2020PA004115, 2021.

Peral, M., Daëron, M., Blamart, D., Bassinot, F., Dewilde, F., Smialkowski, N., Isguder, G., Bonnin, J., Jorissen, F., Kissel, C., Michel, E., Vázquez Riveiros, N., and Waelbroeck, C.: Updated calibration of the clumped isotope thermometer in planktonic and benthic foraminifera, Geochim. Cosmochim. Acta, 239, 1–16, https://doi.org/10.1016/j.gca.2018.07.016, 2018.

Peral, M., Bassinot, F., Daëron, M., Blamart, D., Bonnin, J., Jorissen, F., Kissel, C., Michel, E., Waelbroeck, C., Rebaubier, H., and Gray, W. R.: On the combination of the planktonic foraminiferal Mg/Ca, clumped ($\Delta 47$) and conventional ($\delta 18O$) stable isotope paleothermometers in palaeoceanographic studies, Geochim. Cosmochim. Acta, 339, 22–34, https://doi.org/10.1016/j.gca.2022.10.030, 2022.

Schöne, B. R.: Arctica islandica (Bivalvia): A unique paleoenvironmental archive of the northern North Atlantic Ocean, Glob. Planet. Change, 111, 199–225, https://doi.org/10.1016/j.gloplacha.2013.09.013, 2013.

de Winter, N. J., Witbaard, R., Kocken, I. J., Müller, I. A., Guo, J., Goudsmit, B., and Ziegler, M.: Temperature Dependence of Clumped Isotopes ($\Delta_{47}$) in Aragonite, Geophys. Res. Lett., 49, https://doi.org/10.1029/2022GL099479, 2022.

---

## Author Response (AR3)

Dear Professor Tina Treude,

In this last version, colors in figure 2 were modified to comply with color blindness preparation rules.

Indeed, all data related to this paper are provided in the supplementary files. We took this opportunity to add clumped isotope analysis reports to the supplementary files. Figure axes and panel labels typesetting was corrected in all figures according to the figure guidelines, and call to figure panels were corrected accordingly in the text file.

With best regards,

Thomas Letulle and co-authors.